# AVA-BENCH: Atomic Visual Ability Benchmark for Vision Foundation Models

## ABSTRACT

The rise of vision foundation models (VFMs) calls for systematic evaluation. A common approach pairs VFMs with large language models (LLMs) as general-purpose heads, followed by evaluation on broad Visual Question Answering (VQA) benchmarks. However, this protocol has two key blind spots: (i) Instruction tuning data may not align with VQA test distributions, meaning a wrong prediction can stem from such data mismatch rather than VFMs' visual shortcomings; (ii) VQA benchmarks often require multiple visual abilities in a single question, making it difficult to determine whether errors arise from the lack of all required abilities or just one key ability. To address these gaps, we introduce AVA-BENCH, the first benchmark that explicitly disentangles 14 Atomic Visual Abilities (AVAs)—foundational skills like localization, depth estimation, and spatial understanding that collectively support complex visual reasoning tasks. By decoupling AVAs and matching training and test distributions within each, AVA-BENCH pinpoints exactly where a VFM excels or falters. Applying AVA-BENCH to leading VFMs thus reveals distinctive "ability fingerprints," turning VFM selection from educated guesswork into principled engineering. Notably, we find that a 0.5B LLM yields similar VFM rankings as a 7B LLM while cutting GPU hours by 8×, enabling more efficient evaluation. By offering a comprehensive and transparent benchmark, we hope AVA-BENCH lays the foundation for the next generation of VFMs.

Figure 1: Vision foundation models (VFMs) trained with different data and objectives are evaluated on the proposed AVA-BENCH to assess their strengths and limitations across atomic visual abilities (AVAs).

## 1 INTRODUCTION

Vision Foundation Models (VFMs), pre-trained on large and diverse datasets, have become central to AI by providing transferable features for a wide range of downstream tasks (Khan et al., 2022; Awais et al., 2025; Chowdhury et al., 2025; Bommasani et al., 2021). The variety of pre-training objectives and supervision signals has led to a proliferation of specialized VFMs—such as DINOv2 (Oquab et al., 2023), CLIP (Radford et al., 2021), and SAM (Kirillov et al., 2023)—each excelling in distinct visual

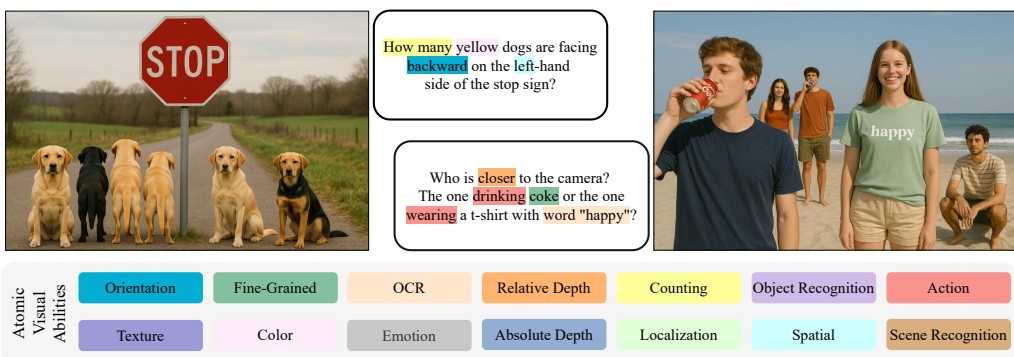

Figure 2: Visual Question Answering (VQA) often requires multiple atomic visual abilities (AVAs) to answer a question. As such, when a model makes an incorrect prediction, it can be difficult to determine whether it stems from a failure to capture all required AVAs or just a single critical one.

capabilities while often exhibiting interesting emergent properties (Goldblum et al., 2023; Caron et al., 2021; Naseer et al., 2021). Consequently, establishing a systematic and effective evaluation protocol for VFMs has become increasingly crucial.

Existing evaluation protocols can generally be categorized into two groups. The first focuses on task-specific capabilities, typically attaching tailored heads to VFMs, followed by fine-tuning and evaluation on dedicated datasets such as ImageNet for classification (Han et al., 2022; Mai et al., 2024) and COCO for detection or segmentation (Thisanke et al., 2023; Balachandran et al., 2024). To better capture the diverse and complex perception challenges of the real world, recent studies advocate a more generic approach that leverages large language models (LLMs) as general-purpose heads, evaluating VFMs on broad Visual Question Answering (VQA) benchmarks (Liu et al., 2023; Zhu et al., 2023; Chowdhery et al., 2023).

While increasingly adopted, this generic protocol may suffer from two potential blind spots: (i) discrepancies between instruction tuning data and VQA test data lead to performance drops due to data mismatch rather than genuine visual limitations in VFMs, and (ii) existing VQA benchmarks typically require multiple visual abilities simultaneously, making it difficult to determine whether a failure arises from the absence of multiple abilities or merely a single critical one.

To address these challenges, we introduce AVA-BENCH, a first VFM evaluation benchmark explicitly designed to disentangle **A**tomic **V**isual **A**bilities (**AVAs**)—the fundamental visual capabilities that combine to solve complex visual reasoning tasks. For instance, answering typical VQA questions shown in Figure 2 necessitates integrating several AVAs.

Specifically, AVA-BENCH evaluates VFMs across **14** carefully identified AVAs (see Figure 3), including **localization**, **counting**, **spatial reasoning**, **orientation**, **absolute and relative depth estimation**, and recognition of **textures**, **colors**, **objects**, **actions**, **emotions**, **optical characters (OCR)**, and **scenes**. Each AVA comes with distribution-matched train and test splits and is probed *in isolation*, eliminating the two aforementioned ambiguities. This enables AVA-BENCH to pinpoint exactly where a VFM excels or falters, providing a clear picture of its strengths and weaknesses.

We systematically benchmark leading VFMs trained under diverse objectives and data (see Figure 1), covering language-supervised (*e.g.*, SigLIP-1/2 (Tschannen et al., 2025; Zhai et al., 2023), CLIP (Radford et al., 2021), InternVL-2.5 (Chen et al., 2024b)), multimodal autoregressive (AIMv2 (Fini et al., 2024)), segmentation-supervised (SAM (Kirillov et al., 2023)), depth-supervised (MiDas (Ranftl et al., 2020)), contrastive self-supervised (DINOv2 (Oquab et al., 2023)), and agglomerative models (RADIO (Ranzinger et al., 2024)). We follow the standard protocol of adding an LLM on top, but fine-tune it separately for each AVA.

Our extensive analyses reveal the following main findings: **(1)** SigLIP-1/2 and AIMv2 emerge as the most versatile VFMs, achieving the highest average rank across all AVAs—*highlighting the critical role of language supervision in enhancing general visual capability*; **(2)** For vision-centric AVAs such as localization, absolute depth estimation, and orientation, the SSL-based DINOv2 performs comparably or better than language-supervised counterparts; **(3)** Conversely, language-centric tasks such as OCR strongly favor language-supervised VFMs; **(4)** Last but not least, we observe that

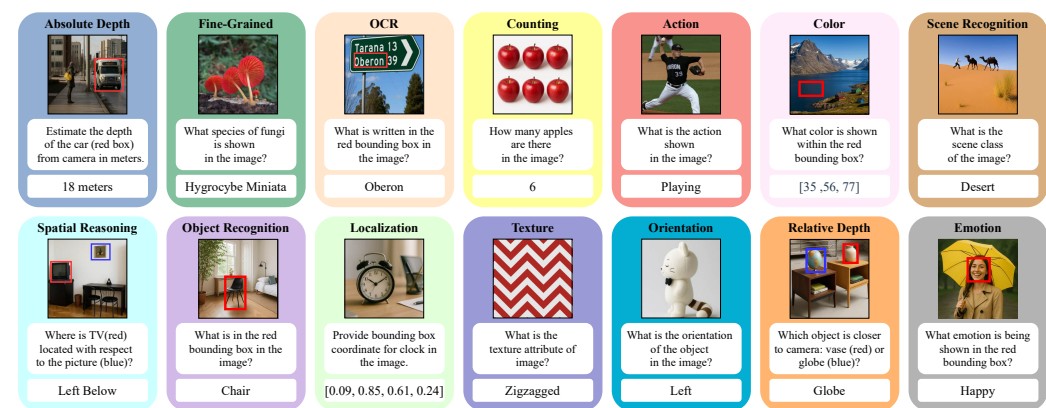

Figure 3: AVA-BENCH consists of 14 Atomic Visual Abilities (AVAs) that can be combined to address more complex visual reasoning tasks.

VFMs universally excel at low- to mid-level AVAs (*e.g.*, texture, relative depth estimation, object recognition), regardless of their training objectives—suggesting that VQA failures typically stem from *deficiencies in specific critical AVAs* rather than broad visual incompetence. These insights turn VFM selection from educated guesswork into principled engineering, enabling practitioners to choose (or ensemble) VFMs based on the specific AVA strengths required by their downstream tasks.

Alongside our main study, we identify a more resource-efficient evaluation strategy. Existing LLM-based VFM evaluations typically rely on heavyweight models such as Vicuna-1.5(7B/13B) (Tong et al., 2024a; Huang & Zhang, 2024), aiming for high absolute accuracy but incurring significant computational overhead. However, when the goal is to compare VFMs, we advocate *prioritizing relative performance over absolute accuracy*. We demonstrate that a lightweight 0.5B LLM preserves reliable VFM rankings while reducing evaluation costs by **8×**, making large-scale analysis substantially more practical. We contextualize our findings within related work (Goldblum et al., 2023; Tong et al., 2024a) to offer a holistic understanding of VFMs and highlight future research directions (section 6).

**Contributions.** Our key contributions are three-fold:

- We identify critical blind spots in existing evaluation protocols and introduce AVA-BENCH, a systematic, diagnostic, and comprehensive VFM evaluation benchmark covering 14 atomic visual abilities (AVAs), clearly highlighting VFMs' fundamental strengths and weaknesses.
- We conduct a detailed evaluation and insightful analysis of diverse leading VFMs, deriving actionable guidance for VFM selection in downstream applications such as customized MLLMs.
- We release a resource-efficient evaluation protocol along with an open-source codebase to facilitate the development of the next generation of accountable and versatile VFMs.

**Remark.** We want to emphasize that AVA-BENCH is designed to systematically evaluate VFMs (*e.g.*, DINOv2) instead of MLLMs (*e.g.*, LLaVA). MLLM in our study solely acts as a general-purpose *prediction head* for the underlying VFM, enabling a unified evaluation interface across tasks.

## 2 VISION FOUNDATION MODELS AND EVALUATION

### 2.1 LLM-BASED EVALUATION FOR VFMS

A wide range of architectures and learning strategies have been explored to develop vision foundation models (VFMs). In general, they fall into two categories based on the nature of their training data. On one hand, VFMs are pre-trained purely on visual data. For instance, the Vision Transformer (ViT) (Dosovitskiy et al., 2020) is trained on labeled images using the supervised signal to capture visual representations. DINOv2 (Oquab et al., 2023), in contrast, adopts a self-supervised learning approach, enabling the model to learn those features without labeling. Some VFMs are designed for specific vision tasks—for example, SAM (Kirillov et al., 2023) specializes in open-vocabulary segmentation, while MiDaS (Ranftl et al., 2020) focuses on monocular depth estimation. RADIO

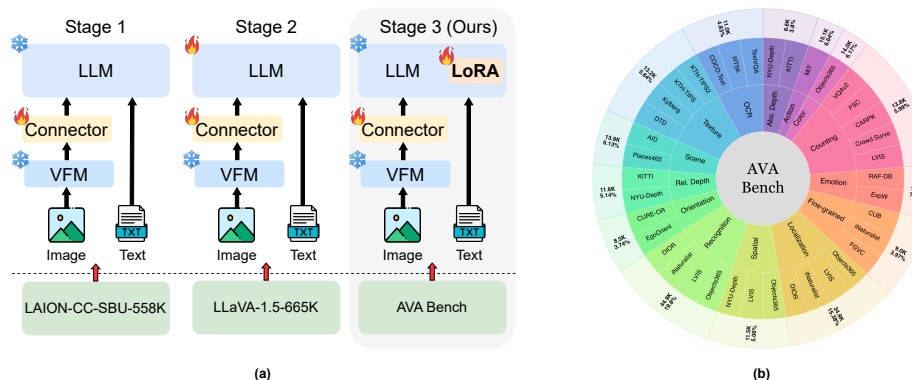

Figure 4: **(a)** Evaluation pipeline for AVA-BENCH: The standard LLaVA-style two-stage training prepares the connector and LLM for VFM evaluation. For each AVA, only connector and LoRA is trained. **(b)** Overall statistics of AVA-BENCH.

(Ranzinger et al., 2024) instead introduces a multi-teacher distillation framework that unifies the strengths of different VFMs (*e.g.*, CLIP, DINOv2, and SAM) into a single efficient student model.

Another family of VFMs leverages image-text pairs. CLIP (Radford et al., 2021) uses contrastive learning to align image and textual descriptions, while SigLIP (Tschannen et al., 2025; Zhai et al., 2023) replaces the contrastive loss with a sigmoid one for more efficient training. Unlike the conventional language-guided VFMs that process images and text separately, AIMv2 (Fini et al., 2024) integrates visual and textual understanding into a single auto-regressive framework, which is simple yet effective. Due to their different architectures and learning objectives, VFMs may have varying strengths and limitations in visual understanding.

## 2.2 LLM-BASED EVALUATION FOR VFMS

Unlike the traditional paradigm, where different perception tasks (*e.g.* classification and segmentation) necessitate task-specific models (Han et al., 2022; Awais et al., 2025), the widespread adoption of Large Language Models (LLMs) as versatile interfaces has significantly shifted this paradigm (Zhang et al., 2024a; Wu et al., 2023). Reflecting this shift, recent studies advocate leveraging large language models (LLMs) as general-purpose heads and evaluating VFMs on broad Visual Question Answering (VQA) benchmarks (Tong et al., 2024a). Specifically, following the LLaVA approach, these studies utilize a two-stage framework: (i) pre-training a connector between frozen LLM and VFMs with image-text pairs for feature alignment (ii) fine-tuning both the connector and the LLM using instruction-tuning data, while keeping VFMs frozen during both stages (see Figure 4 (a)). This LLM-based evaluation protocol has rapidly gained popularity as it closely mirrors the contemporary multimodal LLM setting and effectively captures diverse real-world perception challenges.

However, this evaluation protocol suffers from two potential blind spots. First, discrepancies may exist between instruction-tuning datasets and the test VQA datasets. Thus, a miss-prediction might arise from data mismatch rather than genuine visual deficiencies in VFMs. Second, typical VQA benchmarks typically require multiple visual abilities simultaneously to produce correct answers. This multi-ability requirement obscures the exact cause behind a model's incorrect predictions. For instance, if a VFM is proficient in almost all visual abilities except orientation recognition (forward or backward), it would fail on the question in Figure 2 (left). In such cases, current evaluations only provide relative comparisons between VFMs without elucidating the specific missing capabilities. Thus, AVA-BENCH aims to complement the LLM-based evaluation by pinpointing exactly where a VFM excels or falters, yielding a holistic understanding of their strengths and shortcomings.

## 3 AVA-BENCH

### 3.1 **A**TOMIC **V**ISUAL **A**BILITIES (**AVA**S)

To address the blind spots discussed earlier, we introduce AVA-BENCH, the first systematic evaluation suite explicitly designed to disentangle 14 fundamental perceptual skills–**A**tomic **V**isual **A**bilities

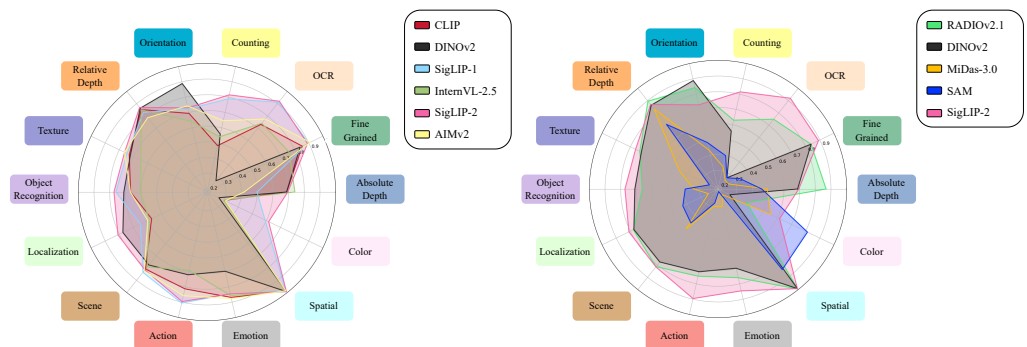

Figure 5: Performance comparison of VFMs across all AVAs. (Left) Language-Supervised VFMs with DINOv2 as a reference. (Right) Other VFMs with the SigLIP-2 as a reference.

(**AVAs**)–for VFMs. AVAs are fundamental perceptual capabilities that can be combined to address more complex visual reasoning tasks. Rather than treating VQA as a monolithic task, we break it down into 14 fundamental perceptual abilities–such as counting, depth estimation, localization and spatial reasoning–each of which can be composed to answer more complex questions. For example, answering the question, *"How many yellow dogs are facing backward on the left-hand side of the stop sign?"*, requires multiple AVAs: counting, color recognition, localization, and spatial reasoning (see Figure 2). By explicitly defining the set of AVAs required to interpret a question, our benchmark quantitatively characterizes where a VFM excels or falters across each core ability.

The AVA selection is grounded in a thorough literature analysis, including compositional text-to-images benchmarks (Huang et al., 2023; Kil et al., 2024; Wu et al., 2024b) and VQA questions (Goyal et al., 2017b; Ainslie et al., 2023) (details in Appendix), focusing strictly on pure perceptual tasks and excluding non-perceptual reasoning skills (*e.g.*, mathematical reasoning). Examples of each AVA can be found in Figure 3 and detailed definitions can be found in Appendix.

## 3.2 DATASET CURATION

Constructing AVA-BENCH required carefully isolating image–question pairs that specifically test individual AVAs. Existing MLLM and VQA datasets often blend multiple perceptual abilities, complicating direct assessments (Tong et al., 2024b; Zhang et al., 2024b; Yu et al., 2023). To achieve clear isolation, we assembled a comprehensive suite comprising image–question pairs from **26** diverse datasets, explicitly aligning each pair with a single targeted AVA. These datasets span a broad range of domains—including general scenes, wildlife (e.g., birds, fungi, plants), vehicles, indoor/outdoor settings, and remote-sensing imagery.

image–question pairs in AVA-BENCH were carefully designed or adapted to focus solely on that AVA. For instance, the question *"What's the depth of the car from the camera?"* involves both localization and depth estimation. To isolate depth estimation, we explicitly provide the car's bounding box. This approach enables fine-grained diagnostic insights into VFMs' AVA-specific strengths and weaknesses. Examples are illustrated in Figure 3, with details below and in the Appendix.

### 3.2.1 LOCALIZATION

We curate **34.8K** localization-focused image–question pairs sourced from: **Objects365** (Shao et al., 2019) and **LVIS** (Gupta et al., 2019) (open-domain), **iNat** (Van Horn et al., 2021) (birds and animals), and **DIOR** (Li et al., 2020) (remote-sensing). To ensure clarity, we include images with a single instance of the target object and exclude objects with small bounding boxes.

### 3.2.2 COUNTING

We collect **13.6K** pairs from: **VQAv2** (Goyal et al., 2017b) (open-domain), **FSC** (Ranjan et al., 2021) (open-domain counting), **CARPK** (Hsieh et al., 2017) (cars), **Crowd Surveillance Dataset** (Li et al., 2022) (people), and **LVIS** (Gupta et al., 2019) (open-domain). FSC, CARPK, and Crowd datasets explicitly cater to counting tasks, ensuring clear and distinct instance counts. LVIS instance masks are utilized to generate counting questions, complemented by counting-focused pairs from VQAv2.

### 3.2.3 SPATIAL REASONING

We curate **11.5K** pairs from **NYU-Depth V2** (Silberman et al., 2012) for indoor scenes, and **LVIS** (Gupta et al., 2019) and **Objects365** (Shao et al., 2019) for open-domain scenarios. For each image, we select two non-overlapping objects—one marked with a blue bounding box (the reference object) and another with a red bounding box (the target). The model is then asked to determine the relative spatial position of the target object with respect to the reference one, choosing from: *top-left*, *top-right*, *bottom-left*, and *bottom-right*.

### 3.2.4 VISUAL ATTRIBUTE

**Orientation.** We compile **8.5K** pairs from two specialized datasets, **CURE-OR** (Temel et al., 2018) and **EgoOrientBench** (Jung et al., 2024), providing uncluttered images of objects from nine distinct orientations: *front*, *back*, *left*, *right*, *top*, *front left*, *front right*, *back left*, and *back right*.

**Color.** We curate **14K** pair sourcing from from open-domain datasets, such as **Objects365** (Shao et al., 2019) and **LVIS** (Gupta et al., 2019). To isolate the color recognition from object semantics or background clutter, we provide bounding boxes that tightly localize regions with minimal color variance, ensuring that the model focuses on RGB prediction.

**Texture.** We curate **13.2K** pairs from: **DTD** (Cimpoi et al., 2014), **Kylberg** (Kylberg, 2011), **KTH-TIPS** and **KTH-TIPS2** (Mallikarjuna et al., 2006) with diverse texture types, such as *striped*, *aluminum foil*, and *zigzagged*.

**Emotion.** We gather **17.0K** pairs from**RAF-DB** (Li & Deng, 2019) and **ExpW** (Lian et al., 2020), covering 7 annotated human emotional states: *surprise*, *neutral*, *disgust*, *fear*, *happy*, *sad*, and *angry*.

### 3.2.5 DEPTH ESTIMATION

We curate depth-focused image-question pairs from **NYU-Depth V2** (Silberman et al., 2012) for indoor scenes and **KITTI** (Geiger et al., 2013) for outdoor scenes.

**Absolute Depth.** We assemble **9K** pairs and for each sample, we place a bounding box on a target object and ask the model to estimate its distance from the camera. We ensure that each object class spans at least five distinct depth bins and that the samples within each bin are balanced.

**Relative Depth.** We collect **11.5K** pairs. Each image is annotated with two non-overlapping bounding boxes for two distinct objects. The model is asked to determine which object is closer to the camera. To prevent annotation bias, we ensure that each object class is evenly distributed between being the nearer or farther object to avoid cases where certain objects are always closer.

### 3.2.6 RECOGNITION

**Action.** We construct **15K** pairs from the **Moments in Time** (Monfort et al., 2019), a video dataset with diverse human actions. We extract the middle frame from each video for 302 distinct actions

**Fine-grained.** We curate **9K** pairs from: **CUB-200** (Wah et al., 2011) (birds), **iNat-21** (Van Horn et al., 2021) (fungi, plants, animals), and **Aircraft** (Maji et al., 2013) (objects). We randomly select 50 species from iNat, 100 species from CUB and all aircraft classes, resulting in a total of **300** classes.

**Object.** We curate **44.9K** pairs from 4 datasets with diverse domains across 70 unique objects: **Objects365** (Shao et al., 2019) and **LVIS** (Gupta et al., 2019) (open-domain), **iNaturalist-2021** (Van Horn et al., 2021) (birds and animals), and **DIOR** (Li et al., 2020) (remote sensing). For each image, we provide a bounding box to eliminate the need for localization.

**Scene.** We curate **13.9K** pairs from two datasets: **Places434** (Zhou et al., 2017) (open-domain) and **AID** (Xia et al., 2017) (remote sensing). The model is required to select the correct scene class from 30 randomly sampled options. The full set includes **464** classes spanning a wide range of scenes.

### 3.3 QUALITY CONTROL AND DATASET STATISTICS

We emphasize rigorous quality control to ensure fair, balanced, and unbiased assessments. Every AVA follows an 80/20 split in which the exact object classes and answer bins that appear in training

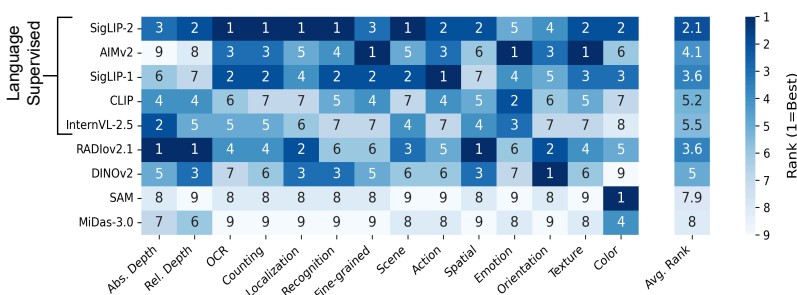

Figure 6: The ranks of VFMs across all AVAs. Detailed results in Appendix.

are mirrored in testing. This ensures that performance differences truly reflect the VFM's perceptual capabilities rather than train-test distribution mismatches, effectively addressing the concerns highlighted in subsection 2.2. Moreover, we try to avoid potential bias in the benchmark. For example, in counting, we explicitly balance the number of samples per counting bin and per object type in both training and testing sets. This approach avoids biases wherein certain counts (e.g., "7 apples") might dominate the training data, artificially inflating accuracy for specific numerical predictions. In another example, for localization, we set a clear threshold for minimum bounding-box area to ensure object visibility. More details about how we ensure the quality of AVA-BENCH can be found in Appendix.

To improve the generalizability within each AVA, we aggregate data from diverse datasets. This ensures that models encounter the same core visual ability across varied scenes, object types, and data distributions, making the assessment more robust and less dataset-specific. Image-question pairs are intentionally crafted or selected to be simple, clear, and explicitly focused on testing only **one** AVA at a time. In summary, AVA-BENCH comprises **218K** meticulously curated image-question pairs that robustly isolate individual AVAs, carefully control dataset balance, visibility, and systematically prevent annotation biases. Statistics are provided in Figure 4 (b), with more details in Appendix.

## 4 EVALUATION PIPELINE OF AVA-BENCH

To evaluate VFMs using AVA-BENCH, we adopt the established LLM-based VFM evaluation protocol, employing the standard LLaVA-style two-stage training procedure to prepare the connector and LLM for VFM evaluation (details in subsection 2.2 and Figure 4 (a)). For each AVA in AVA-BENCH, we fine-tune the connector and LLM while keeping the VFM frozen. Given the modest size of the training sets per AVA (typically around 6K–10K), we employ Parameter-Efficient Fine-Tuning (PEFT) (Mai et al., 2025; Houlsby et al., 2019; Tu et al., 2023b), specifically Low-Rank Adaptation (LoRA) (Hu et al., 2022), to mitigate potential overfitting. Subsequently, the fine-tuned model is evaluated on the corresponding AVA-specific test sets.

### 4.1 IS A HEAVYWEIGHT LLM EVALUATOR NECESSARY?

As discussed in subsection 2.2, traditional LLM-based VFM evaluations, following the LLaVA protocol, predominantly rely on heavyweight models such as Vicuna-1.5 (7B/13B)(Liu et al., 2023)., aiming for high absolute accuracy but incurring considerable computational costs. Nevertheless, a heavyweight LLM may not be mandatory for reliable comparative evaluations. When the goal is to compare VFMs, we advocate *prioritizing relative performance over absolute metrics*. As depicted in Figure 7, a significantly smaller 0.5B LLM (Qwen2) achieves similar relative VFM rankings comparable to a 7B Vicuna-1.5, while dramatically reducing evaluation costs by approximately **8×** (additional details in the Appendix), making large-scale analysis substantially more practical. Thus, we utilize the lightweight 0.5B LLM for all subsequent experiments.

## 5 EXPERIMENTS

All experiment setups follow section 4, with details in Appendix.

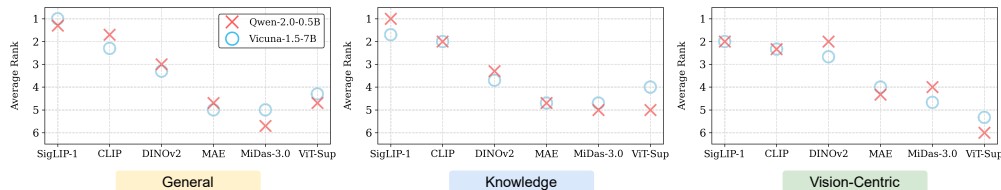

Figure 7: A much smaller 0.5B LLM achieves similar relative VFM rankings comparable to a 7B LLM, while dramatically reducing evaluation costs by approximately **8×**.

**Metrics for AVAs.** For absolute depth and counting AVAs, we utilized a normalized mean absolute error (MAE) relative to the ground-truth. This normalization ensures that errors involving greater distances or counts, which are inherently more challenging, are proportionally penalized less severely. Localization performance is evaluated using Generalized Intersection-over-Union (GIoU (Rezatofighi et al., 2019)), color via CIEDE2000 (Luo et al., 2001), and OCR through Average Normalized Levenshtein Similarity (ANLS) (Biten et al., 2019). All other AVAs employ standard accuracy metrics. Please refer to Appendix for more metric details.

## 5.1 OBSERVATIONS AND ANALYSES

**Bounding boxes isolate and assess specific AVAs.** Bounding boxes serve as a crucial tool for disentangling AVAs, allowing us to isolate and evaluate specific AVAs rather than compounded performance on complex tasks. For example, VFMs exhibit uniformly strong and similar performance in spatial reasoning when provided with ground-truth bounding boxes, indicating a shared strong spatial ability. However, removing the bounding boxes transforms this into a composite task requiring localization and spatial reasoning. As shown in Figure 8(a), such removal leads to substantial divergence in model performance, with rankings closely mirroring their localization ability. This contrast confirms that performance degradation on composite tasks is often attributable to deficiencies in specific underdeveloped AVAs rather than a fundamental failure in all visual capacities.

**Subgroup analyses reveal exceptions hidden by aggregate metrics.** Aggregate metrics can sometimes obscure nuanced insights. To gain a deeper understanding, we conduct detailed analyses by partitioning test samples based on specific criteria (e.g., object size in localization tasks) and examining whether these subgroup trends align with overall performance. We split localization testing samples based on normalized bounding box sizes (relative to image size), where 0.1 indicates an object occupies 10% of the image area. As illustrated in Figure 8 (b), VFMs surprisingly exhibit minimal performance differences when localizing large objects (0.3–0.5). Conversely, performance disparities amplify as object size decreases, revealing significant weaknesses in MiDas and SAM for smaller objects. More subgroup analyses for other AVAs can be found in the Appendix.

**Niche mastery of a modest model.** Interestingly, almost all VFMs, including those with generally lower performance, excel in at least one specific AVA (Figure 6). For example, SAM achieves exceptional results in color recognition, and DINOv2 excels notably in orientation.

**Consistently good VFM performance across lower and mid-level AVAs.** All VFMs, regardless of their training strategies, demonstrate good performance in low- to mid-level AVAs such as texture recognition, relative depth estimation, and object recognition ( Figure 5). This uniformity implies that failures in complex visual reasoning predominantly arise from deficiencies in specific, critical AVAs rather than general shortcomings in visual understanding.

**Language-aligned pretraining boosts performance on language-centric AVAs.** AVAs that involve understanding visual information intertwined with text, such as OCR, substantially benefit from language-aligned pretraining. Non-language-aligned VFMs significantly underperform in these tasks, as illustrated in Figure 5, highlighting the importance of language alignment in VFM training.

**The role of language supervision in VFM.** Language-supervised VFMs, specifically SigLIP-1/2 and AIMv2, demonstrate broad competency across AVAs (Figure 6 & Figure 5). Their consistent high ranking highlights that language supervision is key to developing general-purpose visual abilities.

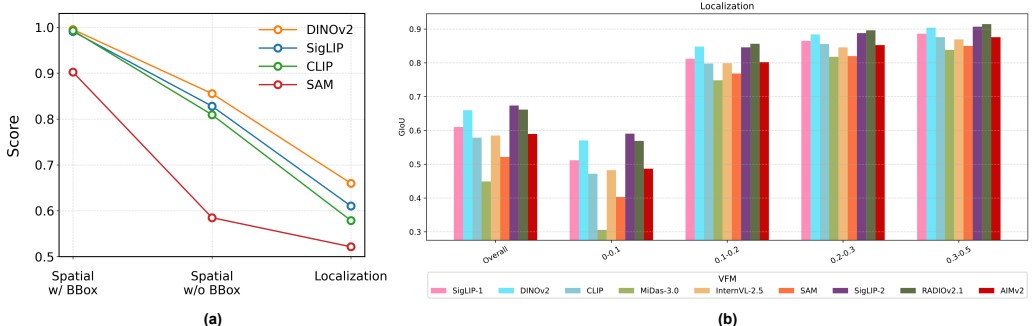

Figure 8: **(a)** Impact of bounding boxes on spatial reasoning performance. With bounding boxes, all VFMs perform perfectly in spatial AVA; without them, models with weaker localization (MiDaS, SAM) perform worse. **(b)** Localization results for different splits based on ground-truth's bounding box sizes. 0.1 means the bounding box size is 10% of the image size. Higher GIoU is better.

## 6 DISCUSSION

**Struggles of Non-Language-Aligned VFMs.** Figure 6 and Figure 5 indicate that non-language-aligned VFMs, despite exhibiting strengths in certain vision-centric or low-to-mid-level AVAs (e.g., DINOv2 in orientation and SAM in color), typically underperform in most AVAs. To investigate the underlying cause of these shortcomings, we conducted a preliminary study on visual feature representations before and after the LLM connector. Using linear probing on max-pooled visual features from DINOv2, we observed a substantial accuracy degradation from 66.3% (pre-connector) to 25.67% (post-connector) for the fine-grained recognition AVA, aligning with findings in previous work (Kim & Ji, 2024). This sharp performance drop suggests that critical visual information is often compromised during modality alignment processes. Recent studies have shown that fine-tuning the last few layers of VFMs can enhance performance (Chen et al., 2024a); however, this approach risks eroding the generalizability that initially makes these VFMs valuable. Alternatively, agglomerative models (Ranzinger et al., 2024; Heinrich et al., 2024) such as RADIO-2.1 demonstrate relatively robust performance on AVA-BENCH, suggesting potential in combining specialized VFMs. Nevertheless, effectively aligning non-language-aligned VFMs to language modalities without sacrificing their inherent visual strengths remains a challenge, highlighting an important avenue for future research.

**Platonic Representation Hypothesis Holds?** Recent research suggests that large-scale training may lead VFMs toward converging onto similar representations–Platonic Representation Hypothesis (Huh et al., 2024). Our findings partially support this hypothesis: for certain AVAs (*e.g.*, object recognition, texture, and relative depth estimation), VFMs demonstrate universally strong performance, indicative of similar underlying visual representations irrespective of training objectives. Conversely, significant performance disparities among VFMs in other AVAs, indicate limitations to the generality of this hypothesis. Thus, our study suggests that this hypothesis might only hold in certain cases and warrants further empirical scrutiny in more diverse and challenging contexts.

**What Are Things Going From Here?** While MLLM have demonstrated remarkable versatility, they are not universally effective in all scenarios, especially in specialized domains (Cheng et al., 2024; Liang et al., 2024). Thus, there is a growing necessity for developing specialized MLLMs (Kumar et al., 2024; Li et al., 2025b). Currently, selecting appropriate VFMs for such customized MLLMs remains largely heuristic (Tong et al., 2024a; Sun et al., 2025; Li et al., 2025a). Our work provides actionable insights that transform this selection process from heuristic guesswork into principled engineering. By clearly identifying AVA-specific strengths and weaknesses, practitioners can now systematically choose VFMs to precisely address the particular visual demands of targeted downstream tasks. Moreover, AVA-BENCH represents a critical step towards developing next-generation VFMs by providing a systematic, diagnostic, and comprehensive evaluation framework. This benchmark enables VFM developers to accurately pinpoint specific deficiencies and implement targeted improvements, fostering the creation of more robust, versatile, and well-rounded VFMs in the future.

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

## APPENDIX

**Disclosure of LLM Usage.** Portions of this manuscript were polished for clarity and readability using an LLM. The LLM was not used to generate research ideas, design experiments, analyze data, or draw conclusions. All scientific content, methods, and results are the authors' original work.

We provide details omitted in the main paper.

- Appendix A : Example and curation details of each AVA of AVA-BENCH
- Appendix B : Details of hyperparameter and metrics used in experiments
- Appendix C: Additional results and detailed analysis
- Appendix D: Detailed overview of VFMs
- Appendix E: Related work
- Appendix F: Evaluation of efficiency
- Appendix G Dataset copyright/license

## A   AVA-BENCH DETAILS

| Atomic Visual Ability | Definition | Example Question |
|---|---|---|
| Counting | Determining the number of instances of an object | How many apples are in the image? |
| Localization | Identifying the location of an object in the image | Provide bounding box coordinate for bicycle. |
| Fine-Grained | Differentiating between similar sub-categories of objects | What species of fungi is in the image? |
| OCR | Reading and interpreting text visible in the image | What is written in the red bounding box in the image? |
| Absolute Depth | Estimating how far an object is from the camera | From the camera's perspective, estimate how far the closest point of the car (red box) is from the camera in real-world distance, in meters. |
| Relative Depth | Comparing distances of two objects from the camera | Which object is closer to the camera, the car (red box) or the cyclist (blue box) to the camera? |
| Orientation | Determining the facing direction or angle of an object | What is the orientation of the toy bus in the image? |
| Spatial | Inferring layout and spatial relations | Considering the relative positions of two objects in the image, where is the bicycle (red box) located with respect to the towel (blue box)? |
| Object Recognition | Identifying objects present in the image given bounding box | What is in the red bounding box in the image? |
| Scene Recognition | Identifying the broader environment or type of setting | What is the scene class of the image? |
| Action Recognition | Determining what action is being performed | Which action or activity is shown in the image? |
| Texture | Describing surface appearance or material of objects | What is the texture attribute of image? |
| Color | Identifying colors of objects given bounding box | What color is shown within the bounding box? |
| Emotion | Recognizing emotional expressions in humans given bounding box | What emotion is being shown in the image? |

Table 1: **Atomic Visual Abilities (AVAs).** We identify 14 AVAs, serving as the foundational capabilities that can be combined to tackle complex visual reasoning tasks. For each AVA, we provide the definition and an example question in `ourbench`.

## A.1  ATOMIC VISUAL ABILITIES (AVAS)

As mentioned in Section 3.1, AVAs are elemental visual capabilities that can be combined to address more complex visual reasoning tasks. The definitions and representative questions for each AVA can be found in Table 1. Additional qualitative illustrations are provided in Figure 2 and Figure 4 (b).

The 14 AVAs selected for AVA-BENCH are grounded in a thorough literature analysis.

1. **Compositional text-to-image (T2I) benchmarks.** Studies on controllable generation motivate core visual primitives—number, colour, texture, object identity, spatial relations, and more—used to construct compositional prompts (Huang et al., 2023; Wu et al., 2024b). These primitives form an initial pool of candidate abilities.

2. **VQA question analysis.** We employ GPT-4 to summarize the visual skills demanded by VQA questions in various commonly-used datasets (VQAv2 (Goyal et al., 2017b), RealWorldQA (xAI, 2024), GQA (Ainslie et al., 2023), etc.), thereby enriching the pool with abilities emphasized by real-world questions.

Intersecting these two sources yields a concise yet crucial set of AVAs. Moreover, we focus strictly on pure perceptual tasks and exclude non-perceptual reasoning skills (e.g., historical context, mathematical reasoning). We provide more related work discussion in Appendix E.

## A.2  DATASET CURATION

**Spatial Reasoning (Wu et al., 2025a; Zhang et al., 2025a).** We curate **11.5K** image pairs from **NYU-Depth V2** (Silberman et al., 2012) (indoor scenes) and **LVIS** (Gupta et al., 2019) and **Objects365** (Shao et al., 2019) (open-domain scenes), all containing instance segmentation annotations. In each image, two distinct, non-overlapping objects are selected—one highlighted with a blue bounding box (reference object) and another with a red bounding box (target object). The model must identify the relative spatial position of the red box with respect to the blue box, choosing from four multiple-choice options: *Left above*, *Left below*, *Right above*, and *Right below* (Figure 9). The preprocessing steps for dataset creation are summarized below:

- To prevent ambiguity in interpretation, we restrict each image to contain only one instance of the target and reference objects.

- Object pairs whose bounding boxes overlapped either horizontally or vertically were excluded, ensuring unambiguous assignment to the four spatial categories.

- Extremely small bounding boxes complicating localization were filtered out. Specifically, object instances covering at least 2% of the image area for NYU-Depth V2, and at least 0.2% for LVIS and Objects365, were retained.

- For every question, we have a target and a reference object with a spatial position (the relative position of the target based on the reference object). Each object class was ensured to appear in multiple spatial positions, with 40 samples per spatial position category. For each target object class in each spatial position, we ensure diversity of reference by selecting samples from 8 distinct reference classes and drawing 5 rows per reference, thereby preventing overfitting to a small set of co-occurring anchors.

- Each object class was ensured to appear in multiple spatial positions, with 40 samples per spatial position category. This prevents models from memorizing fixed layouts.

- For each target object class, reference object class, and spatial position category, an 80% train and 20% test split was ensured, for uniform distribution and fair evaluation.

- The question for each pair: *"Considering the relative positions of two objects in the image, where is the microphone (annotated by the red box) located with respect to the speaker (annotated by the blue box)? Choose from A. Left above, B. Left below, C. Right above, D. Right below."*

**Counting (Yao et al., 2025; Li et al., 2024a).** We curate a total of **13.6K** images from five datasets— **VQAv2** (Goyal et al., 2017b), **FSC-147** (Ranjan et al., 2021), **CARPK** (Hsieh et al., 2017), **LVIS** (Gupta et al., 2019) , and **CrowdHuman** (Li et al., 2022) —to evaluate object counting abilities

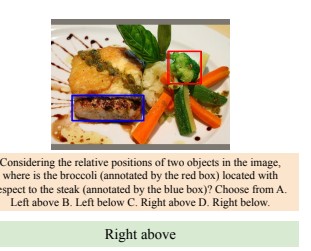
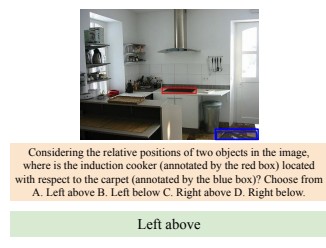

Figure 9: Examples of Spatial Reasoning AVA Samples.

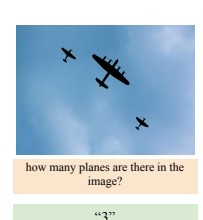
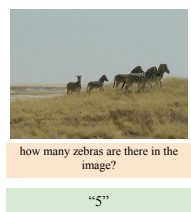
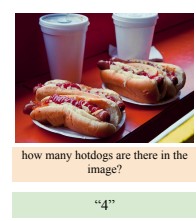
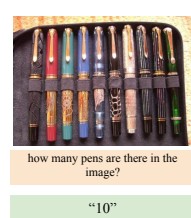

Figure 10: Examples of Counting AVA Samples.

across diverse domains, including open-domain scenes, natural objects, structured environments, and densely crowded contexts. Each image is paired with a question prompting the model to count the number of instances of a specified object category, and the model must return an integer-valued answer(Figure 10). The preprocessing steps for dataset creation are summarized below:

- To ensure valid supervision, we filter all samples to retain only those with non-zero object counts and with object_count $\leq 40$.

- For each object category (object id), we require at least 4–5 distinct object count values to be represented, preventing overfitting to static object layouts.

- For each object count, we sample a fixed range of images per object_id—between 6 and 30 depending on the dataset—to balance frequency and diversity.

- Dataset-specific sampling rules are applied:

  - VQAv2 and LVIS: 15–30 images per object count; $\geq 5$ count values per object id.
  - FSC-147: 6–12 images per object count; $\geq 4$ count values per object id.
  - CARPK: 10–20 images per object count; $\geq 5$ count values per object id.
  - CrowdHuman: object count capped at 40; 25–50 images per count level.

- An 80% train / 20% test split is maintained for each object id and count level to ensure balanced distribution during evaluation.

- The question for each image is: *"How many [object] are there in the image?"*, where [object] refers to the annotated target category.

**Fine-grained (Paul et al., 2023; Gu et al., 2025; Zhang et al., 2025c).** We curate a total of **9K images** from five fine-grained recognition domains—**Bird**, **Animal**, **Fungi**, **Plant**, and **Object**—to assess species-level recognition capabilities. The dataset sources include CUB-200-2011 (Wah et al., 2011) for birds, iNat21 (Van Horn et al., 2021) for animals, fungi, and plants, and FGVC Aircraft (Maji et al., 2013) for objects. Each sample contains an image and a question prompting the model to identify the specific species or object type(Figure 11). Construction details are as follows:

- We select 100 bird species from CUB-200-2011, and 50 random classes each from the Animal, Fungi, and Plant categories of iNat21, as well as 50 classes from FGVC Aircraft for the Object category. All random selections use a fixed random seed to ensure reproducibility.

- This results in 300 total object ids. For each class, we uniformly sample 30 images.

| Atomic Visual Abilities (AVA) | Dataset | Domain | # Train Samples | # Test Samples |
|---|---|---|---|---|
| Localization | Objects365 (Shao et al., 2019) | Open | 27.9K | 6.9K |
| | LVIS (Gupta et al., 2019) | Open | | |
| | iNaturalist-2021 (Van Horn et al., 2021) | Bird, Animal | | |
| | DIOR (Li et al., 2020) | Remote-Sensing | | |
| Counting | VQAv2 (Goyal et al., 2017b) | Open | 10.8K | 2.8K |
| | FSC (Ranjan et al., 2021) | Open | | |
| | CARPK (Hsieh et al., 2017) | Car | | |
| | Crowd Surveillance (Li et al., 2022) | People | | |
| | LVIS (Gupta et al., 2019) | Open | | |
| Fine-grained | CUB-200-2011 (Wah et al., 2011) | Bird | 7.2K | 1.8K |
| | iNaturalist-2021 (Van Horn et al., 2021) | Fungi,Plant, Animal | | |
| | FGVC-Aircraft (Maji et al., 2013) | Object | | |
| Absolute Depth | NYU-Depth V2 (Silberman et al., 2012) | Indoor Scene | 6.8K | 1.8K |
| | KITTI (Geiger et al., 2013) | Outdoor Scene | | |
| Relative Depth | NYU-Depth V2 (Silberman et al., 2012) | Indoor Scene | 9.2K | 2.4K |
| | KITTI (Geiger et al., 2013) | Outdoor Scene | | |
| OCR | COCO-Text (Veit et al., 2016) | Open | 8.8K | 2.2K |
| | IIIT5K (Mishra et al., 2012) | Open | | |
| | TextVQA (Goyal et al., 2017a) | Open | | |
| Orientation | EgoOrientBench (Jung et al., 2024) | Open | 6.9K | 1.6K |
| | CURE-OR (Temel et al., 2018) | Indoor | | |
| Object Recognition | Objects365 (Shao et al., 2019) | Open | 37.9K | 7K |
| | LVIS (Gupta et al., 2019) | Open | | |
| | iNaturalist-2021 (Van Horn et al., 2021) | Bird, Animal | | |
| | DIOR (Li et al., 2020) | Remote-Sensing | | |
| Action Recognition | MiT (Monfort et al., 2019) | Open | 12K | 3K |
| Texture | DTD (Cimpoi et al., 2014) | Open | 10.6K | 2.7K |
| | Kylberg (Kylberg, 2011) | Open | | |
| | KTH-TIPS (texture classification & segmentation, 2024) | Open | | |
| | KTH-TIPS2 (Mallikarjuna et al., 2006) | Open | | |
| Spatial Reasoning | Objects365 (Shao et al., 2019) | Open | 9.9K | 1.6K |
| | LVIS (Gupta et al., 2019) | Open | | |
| | NYU-Depth V2 (Silberman et al., 2012) | Indoor Scene | | |
| Scene Recognition | Places434 (Zhou et al., 2017) | Open | 11.1K | 2.8K |
| | AID (Xia et al., 2017) | Remote-Sensing | | |
| Emotion | RAF-DB (Li & Deng, 2019) | Human | 11.9K | 5.1K |
| | ExpW (Lian et al., 2020) | Human | | |
| Color | Objects365 (Shao et al., 2019) | Open | 11.2K | 2.8K |
| Total | - | - | 182.2K | 44.5K |

Table 2: **Detailed statistics of** AVA-BENCH.

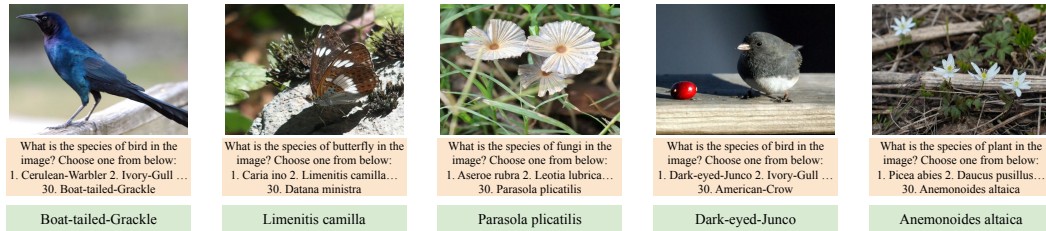

Figure 11: Examples of Fine-grained recognition AVA Samples.

- For iNat21 entries, we format species names by retaining only the last two words of their taxonomic labels for clarity and consistency.

- For each object class, an 80% training and 20% testing split was established to ensure balanced per-class evaluation.

- For each multiple-choice question, the candidate list includes all 50 or 100 (Object only) class names used in that task split, ensuring consistent, closed-set evaluation.

- The question for each image is: *"What species of bird is in the image? Choose one from below: 1. Cerulean_Warbler, 2. American_Crow, ..., 100. Pine_Warbler"*

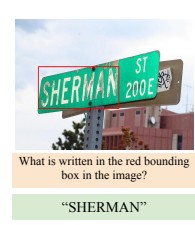 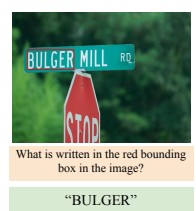 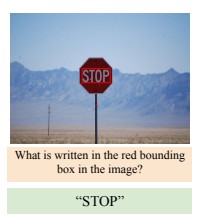 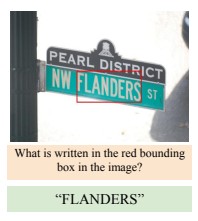 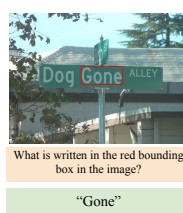

Figure 12: Examples of OCR AVA Samples.

**OCR (Huang et al., 2025; Liu et al., 2024).** We curate **10.9K** images from three OCR datasets—**COCO-Text** (Veit et al., 2016), **IIIT5K** (Mishra et al., 2012), and **TextVQA** (Goyal et al., 2017a)—each containing word-level bounding box annotations. In each image, a red bounding box highlights the word to be transcribed. The model is prompted to recognize the textual content inside the box based on visual context(Figure 12). The preprocessing steps for dataset creation are summarized below:

- For COCO-Text, we retain only word-level boxes with an area greater than 1500 pixels to ensure sufficient visual resolution.
- For TextVQA, we apply a stricter filtering criterion by retaining only word boxes with area larger than 2000 pixels.
- For IIIT5K, we randomly sample 2,000 images from the original dataset without applying any area-based filtering.
- Each dataset is split into 80% training and 20% validation subsets individually, before merging the resulting splits to form the final OCR benchmark.
- A red bounding box is rendered on each image to highlight the target word location during inference.
- The question for each image: *"What is written in the red bounding box in the image?"*

**Localization (Wu et al., 2024a; Sapkota & Karkee, 2025).** We curate **34.8K** localization-focused image–question pairs sourced from **Objects365** (Shao et al., 2019) and **LVIS** (Gupta et al., 2019) (open-domain), **iNaturalist-2021** (Van Horn et al., 2021) (birds and animals), and **DIOR** (Li et al., 2020) (remote-sensing). Each image contains a single object instance, and the model is prompted to identify its location by providing the bounding box coordinates(Figure 13). The preprocessing steps for dataset creation are summarized below:

- We only retain object instances whose category appears exactly once in an image, to avoid ambiguity in localization supervision.
- We filter out objects with extremely small or large bounding boxes, retaining only those whose normalized area falls within the range of $0.002 < \text{area} < 0.5$ relative to the image.
- For **Objects365** and **LVIS**, we manually select 20 target object categories each. If the number of valid images for a category exceeds 700, we randomly sample 700 using a fixed seed for reproducibility.
- For **iNaturalist-2021**, we select 10 categories from the `aves` (birds) class and 10 from the `mammalia` (mammals) class, following the same filtering and sampling strategy. All scientific names are mapped to common names to improve interpretability and model alignment.
- For **DIOR**, we follow the same filtering steps and manually select 10 target categories, sampling up to 700 images per category as needed.
- All images are padded to square format using a consistent background color computed as the mean RGB value across multiple image processors:

$$\text{background color} = RGB(124, 120, 111)$$

The padding preserves content aspect ratio and ensures uniform input dimensions across models.

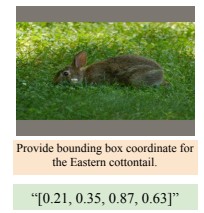 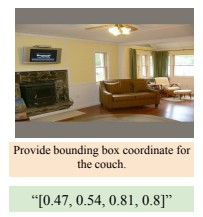 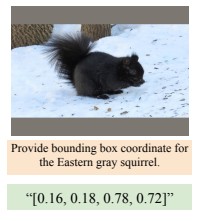 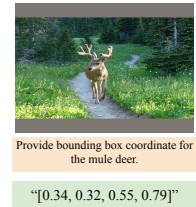

| Provide bounding box coordinate for the song sparrow. | Provide bounding box coordinate for the Eastern cottontail. | Provide bounding box coordinate for the couch. | Provide bounding box coordinate for the Eastern gray squirrel. | Provide bounding box coordinate for the mule deer. |
| "[0.36, 0.25, 0.7, 0.8]" | "[0.21, 0.35, 0.87, 0.63]" | "[0.47, 0.54, 0.81, 0.8]" | "[0.16, 0.18, 0.78, 0.72]" | "[0.34, 0.32, 0.55, 0.79]" |

Figure 13: Examples of Localization AVA Samples.

- For each object category, an 80% training and 20% testing split was performed after filtering and sampling, ensuring balanced and fair evaluation.

- The question for each image follows the format: *"Provide bounding box coordinate for red-tailed hawk."* The object name is dynamically replaced depending on the image.

**Recognition (Fu et al., 2024; Zhang et al., 2025b; Tu et al., 2023a).** We curate **44.9K** recognition samples from four datasets spanning diverse visual domains—**Objects365** (Shao et al., 2019) and **LVIS** (Gupta et al., 2019) (open-domain objects), **iNaturalist-2021** (Van Horn et al., 2021) (birds and animals), and **DIOR** (Li et al., 2020) (remote sensing). These samples are derived from the same images and object instances used in the localization benchmark. However, instead of asking for bounding box prediction, the recognition task requires the model to identify the object within a visually highlighted region (Figure 14).

The preprocessing steps for dataset creation are summarized below:

- We apply the same curation strategy as in localization: only one valid instance per image, with normalized bounding box area between $0.2\%$ and $50\%$ of the image. For each dataset, 10 or 20 object categories are manually selected.

- From **Objects365** and **LVIS**, we select 20 object categories each, and randomly sample up to 700 images per category (using a fixed seed for reproducibility).

- From **iNaturalist-2021**, we retain 10 species from the `Aves` (birds) and 10 from `Mammalia` (mammals) branches. Scientific names are converted to common English names for accessibility. Each species contributes up to 700 images.

- From **DIOR**, we select 10 object categories and apply the same filtering and sampling strategy (max 700 images per class).

- All images are padded to square shape using a consistent background color, computed from the average mean pixel values across nine vision-language processors, to ensure uniform input dimensions.

- For each object category, an 80% training and 20% testing split was performed after filtering and sampling, ensuring balanced and fair evaluation.

- **Unlike localization**, where bounding boxes are not rendered and must be predicted, in recognition the **red bounding box is explicitly drawn** onto each image to guide the model's attention.

- The question format is: *"What is in the red bounding box? Choose from the following option: 1. airport, 2. american robin, ..., 70. vulpes vulpes"* The 70 object categories are shared across datasets and randomly shuffled for each question instance.

**Color (Wang et al., 2025; Chiu et al., 2024).** We curate **14K** images from **Objects365** (Shao et al., 2019) and **LVIS** (Gupta et al., 2019), each contributing 7K samples. This AVA focuses on assessing **color perception** in natural scenes. For each image, we extract a coherent color region using the following pipeline:

- We apply SLIC superpixel segmentation and convert the image to LAB color space.

- Superpixels with similar color values are merged to form larger regions of consistent color.

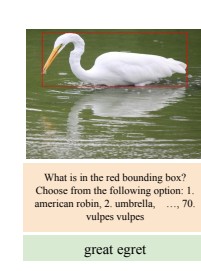 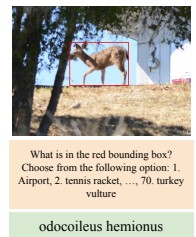 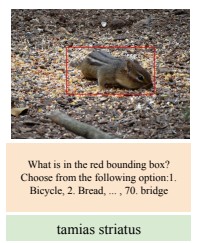 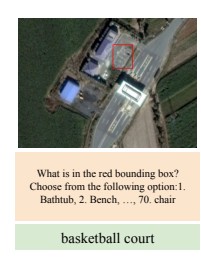 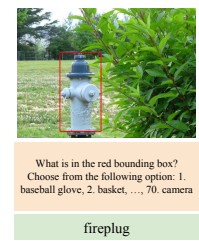

Figure 14: Examples of Recognition AVA Samples.

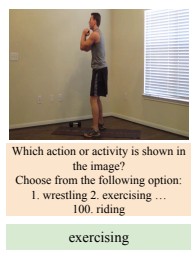 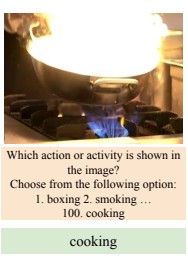 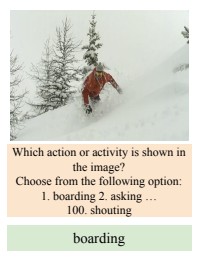 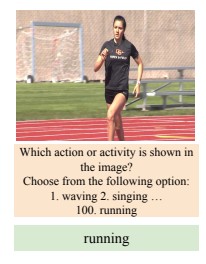 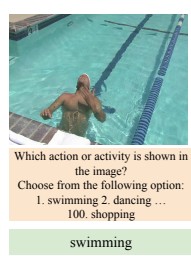

Figure 15: Examples of Action AVA Samples.

- Among all candidate regions, we select the **top-1 region with the lowest internal color variance** as the final choice.

- A red bounding box is drawn on the selected region, and the most frequent RGB color within this region is used as the answer.

- For each object category, an 80% training and 20% testing split was performed after filtering and sampling, ensuring balanced and fair evaluation.

- The question format for each sample is: "What color is shown within the red bounding box?"

**Action (Szot et al., 2024; Wang et al., 2025; Xie et al., 2025).** We curate a total of **15K image–question pairs** from the **Moments in Time** (Monfort et al., 2019) dataset, covering a wide range of human actions and activities. Each sample is derived from a short video clip annotated with a specific action label (Figure 15). Construction details are as follows:

- Similar action labels are merged into a unified category for clarity.

- For each class, we randomly sample up to 500 training videos and 100 validation videos, ensuring broad yet balanced category coverage.

- The middle frame of each selected video is extracted and used as the image representing the associated action.

- Since listing all classes may exceed the token limits of many vision–language models, we randomly sample 100 action options per question, ensuring the ground-truth answer is always included.

- The question format for each sample is: "Which action or activity is shown in the image? Choose from the following option: 1. buying, 2. catching, ..., 100. boxing"

**Emotion (Yang et al., 2024; Li et al., 2024b).** We curate **17K image-question pairs** from two large-scale facial expression datasets: **RAF-DB** (Li & Deng, 2019) and **ExpW** (Lian et al., 2020). These datasets consist of human portraits labeled with one of seven basic emotions: *happy*, *sad*, *angry*, *fear*, *surprise*, *neutral*, and *disgust*. Each image is annotated with a bounding box localizing the face of interest.

The preprocessing steps for dataset creation are summarized below:

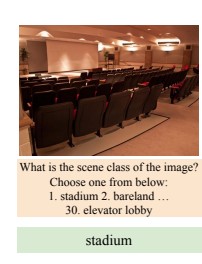 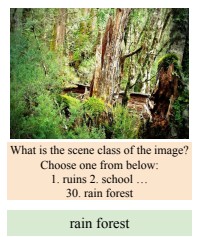 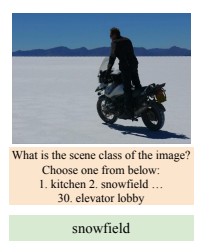 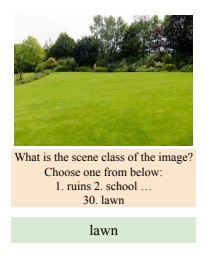 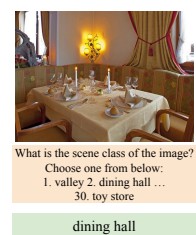

Figure 16: Examples of Scene AVA Samples.

- Emotion labels across datasets were unified by consolidating synonymous terms (e.g., `happiness` and `happy`, `anger` and `angry`) to ensure consistent categorization across all samples.

- Bounding box annotations provided in the datasets were used to highlight the specific individual in multi-person scenes.

- An 80/20 train-test split was applied independently per emotion category to maintain class balance during evaluation.

- The question for each image is framed as: *"Which of the following best describes the person's emotion in the red box? 1. happy, 2. sad, 3. angry, 4. fear, 5. surprise, 6. neutral, 7. disgust."*

**Scene (Dai et al., 2025; Fan et al., 2024).** We curate **13.9K image-question pairs** from two diverse datasets: **Places434** (Zhou et al., 2017) (open-domain) and **AID** (Xia et al., 2017) (remote sensing). Each image is paired with a multiple-choice question, where the model selects the correct scene category from a pool of 30 randomly sampled options. The final set includes **463 unique scene classes** spanning a wide range of environments(Figure 16). The preprocessing steps for dataset creation are summarized below:

- GPT-4o was utilized to standardize labels across datasets by converting fine-grained labels into single, unified labels. Humans carefully checked each conversion to merge the newly converted labels conveying the same meaning with existing labels, ensuring semantic clarity and fluency.

- A uniform distribution was maintained by extracting exactly 30 images per scene category, preventing category imbalance and ensuring consistent representation across classes. Classes with less than 30 images were discarded.

- For each scene category within both datasets, an 80% training and 20% testing split was established, ensuring balanced and fair evaluation conditions.

- The question for each pair: "What is the scene class of the image? Choose one from below: 1. Entrance hall, 2. Lawn, ... 30. Snowy Mountain." These 30 options were selected by randomly sampling from the complete set of scene classes within each respective dataset, maintaining diversity and preventing predictable patterns.

**Texture (Eppel et al., 2025; Gavrikov et al., 2024).** To assess a VFM's ability to distinguish fine-grained visual patterns, we curate **13.2K image-question pairs** from diverse surface textures using close-up images from four open-domain datasets: **DTD** (Cimpoi et al., 2014) , **Kylberg** (Kylberg, 2011) , **KTH-TIPS** (texture classification & segmentation, 2024) and **KTH-TIPS2-b** (Mallikarjuna et al., 2006). These datasets encompass a diverse array of texture types—such as *striped*, *aluminum foil*, and *zigzagged*—capturing subtle visual patterns that are essential for accurate texture recognition. Each image is paired with a multiple-choice question, requiring the model to select the correct texture label from a set of options(Figure 17). The preprocessing steps for dataset creation are summarized below:

- Images where textures appeared as part of larger objects in cluttered scenes or within complex real-world photographs were discarded, to ensure that textures were clearly localized and recognizable without contextual interference.

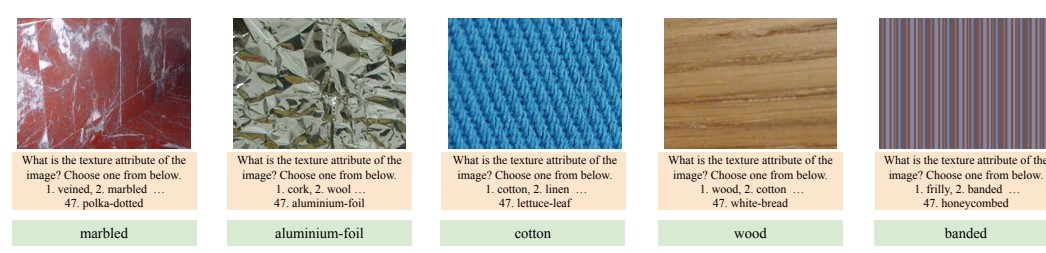

Figure 17: Examples of Texture AVA Samples.

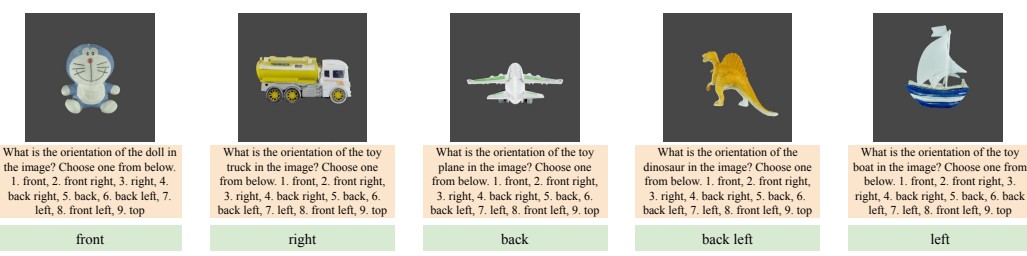

Figure 18: Examples of Orientation AVA Samples.

- Each texture attribute was represented by multiple images, with a minimum of 120 and a maximum of 480 samples per attribute, ensuring diversity and preventing memorization of fixed patterns by the models.

- For each texture attribute, an 80% train and 20% test split was ensured, reaching uniform distribution and fair evaluation.

- The question for each pair:*"What is the texture attribute of the image? Choose one from below: 1. banded, 2. blotchy, . . . , 47. veined."* The provided options exactly match the entire option pool from each respective dataset and were shuffled to avoid bias.

**Orientation (Yin et al., 2025; Jung et al., 2024).** To evaluate viewpoint understanding, we curate **8.5K image-question pairs** from two specialized datasets: **CURE-OR** (Temel et al., 2018) and **EgoOrientBench** (Jung et al., 2024) . These datasets provide uncluttered images of objects captured from nine distinct orientations—*front*, *back*, *left*, *right*, *top*, *front left*, *front right*, *back left*, and *back right*—allowing models to learn pose-specific cues without requiring bounding boxes(Figure 18). The preprocessing steps for dataset creation are summarized below:

- For EgoOrientBench, each object class was ensured to appear in multiple orientations, with at least 10 and at most 40 samples per orientation label. This encourages models to learn generalized representations rather than memorizing specific arrangements.

- From CURE-OR, we selected object instances photographed against two different background conditions using three distinct capture devices, ensuring variation in imaging style without compromising clarity.

- For each object, 80% of the images from each orientation were assigned to the training set, and the remaining 20% to the test set, ensuring balanced representation and fair evaluation across orientations.

- The question for each pair: *"What is the orientation of the toy plane in the image? Choose one from below: 1. front, 2. front right, 3. right, 4. back right, 5. back, 6. back left, 7. left, 8. front left, 9. top"*. These nine options represent the common orientation labels provided by both datasets.

**Absolute Depth (Xia & Wu, 2024; Mi et al., 2024; Zhang & Lu, 2025).** We curate **9K** image-object pairs from **NYU-Depth V2** (Silberman et al., 2012) for indoor scenes and **KITTI** (Geiger et al., 2013) for outdoor scenes. These datasets contain aligned RGB and depth information. Each image includes an object annotated with a bounding box, and the task requires estimating the absolute

depth of the object in meters. This value is then matched against discretized ground-truth bins for evaluation(Figure 19).

The preprocessing steps for **indoor** dataset creation are summarized below:

- Ambiguous object categories were excluded via a manually curated list of label IDs that often lack clear boundaries or meaningful depth interpretations (e.g., wall, floor, ceiling, etc.).

- Images were resized to a fixed resolution of $384 \times 384$, padding vertically as needed to preserve the original aspect ratio.

- A minimum bounding box area threshold was enforced after resizing all images. Specifically, we filtered out bounding boxes smaller than 500 pixels to ensure sufficient spatial resolution for the model.

- To ensure depth variation and avoid trivial samples, only object classes with at least 3 distinct depth bins (i.e., meaningful distribution over depth) were retained.

- For each label, depth bins were required to have a minimum of 10 and a maximum of 30 image samples. We removed bins with insufficient data to meet this requirement and capped those with excess samples by sorting instances based on bounding box area.

- After filtering, we retained 45 object classes, resulting in 4.4K unique image-object pairs. The dataset was split into 80% train and 20% test, preserving label and bin balance.

- The question for each pair: *"From the camera's perspective, estimate how far the closest point of the cabinet (highlighted by a red box) is from the camera in real-world distance, in meters. Select the best answer from the options below: A. 1-2, B. 2-3, C. 3-4, D. 4-5, E. 5-6, F. 6-7".*

The preprocessing steps for **outdoor** dataset creation are summarized below:

- To ensure the depth estimation task remains non-trivial, only objects with a minimum distance of 8 meters from the camera were considered. This avoids bias toward near-field predictions and better evaluates model precision in far-range perception.

- Depth values were discretized into bins, and for each class, we selected between 20 and 60 samples per bin to ensure coverage while avoiding overrepresentation. Bins with fewer than 20 samples were discarded. When bins exceeded 60 samples, selection was sorted by bounding box area to prioritize larger, more reliable objects.

- Image crops were extracted per object while maintaining the following aspect ratio constraint to preserve visual consistency: the width of the crop must be within the range [height, $2 \times$ height]. This was enforced using the original image dimensions before padding or resizing.

- Objects touching any edge of the image were excluded to reduce the likelihood of partial occlusion or clipping.

- Images were padded vertically as needed to preserve the original aspect ratio.

- The final outdoor absolute depth set contains approximately 6K samples. For each object class and depth bin, an 80/20 train-test split was applied to maintain consistency in evaluation.

- The question for each outdoor sample: *"Estimate the distance from the camera to the closest part of the cyclist (highlighted by a red box) in meters. Choose the best option: A. 8-9, B. 10-11,..., H. 30-31."*

**Relative Depth (Xia & Wu, 2024; Mi et al., 2024; Zhang & Lu, 2025).** We curate **11.6K** image-object pairs from **NYU-Depth V2** (Silberman et al., 2012) for indoor scenes and **KITTI** (Geiger et al., 2013) for outdoor scenes, targeting the task of identifying which of two objects in an image is closer to the camera. Each image contains two distinct objects, each annotated with a bounding box. The model is asked to compare their absolute depth and choose the object that appears closer to the camera (Figure 20).

The preprocessing steps for **indoor** dataset creation are summarized below:

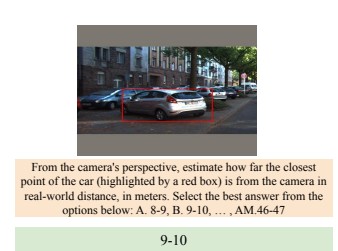 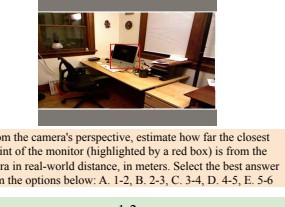 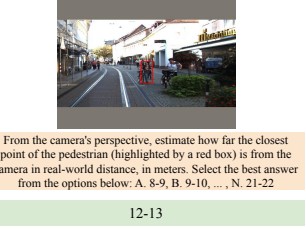

Figure 19: Examples of Absolute Depth AVA Samples.

- Ambiguous object categories were excluded via a manually curated list of label IDs that often lack clear boundaries or meaningful depth interpretations (e.g., wall, floor, ceiling, etc.).

- Only object pairs with valid bounding boxes (i.e., non-overlapping, fully inside image boundaries) were considered.

- To ensure perceptual clarity, candidate object pairs were filtered by requiring an absolute depth difference of at least **0.5 meters** between them.

- For a given object class, only those with at least **10** valid pairings were retained, and a maximum of **30** total pairings per label were allowed.

- After filtering and sampling, we retained **131** object pairs across **5.7K** total questions. Images were padded vertically as needed to preserve their original aspect ratio. These were split into 80% train and 20% test sets while maintaining object label and depth-difference balance.

- Each question is posed as: *"Estimate the real-world distances between the objects in this image. Which object is closer the camera, the sink (highlighted by a red box) or the towel (highlighted by a blue box) to the camera? Choose one option from below: 1. red, 2. blue"*.

The preprocessing steps for **outdoor** dataset creation are summarized below:

- Object annotations were sourced for the following categories: `Car`, `Van`, `Pedestrian` (merged with `Person sitting`), and `Cyclist`.

- Pairs were formed using both intra-class (e.g., Car vs. Car) and inter-class (e.g., Car vs. Pedestrian) combinations.

- Pairs were retained only if the depth difference between the two objects was at least 0.5 meters, ensuring a meaningful perceptual gap.

- To avoid ambiguity and incomplete visual evidence, the following filters were applied:
  - Pairs with occluded objects were excluded.
  - Pairs where either object was touching the image edge were discarded.
  - Only crops satisfying the aspect ratio constraint height $\leq$ width $\leq 2 \times$ height were included, ensuring visual consistency.
  - Images were padded vertically as needed to preserve their original aspect ratio.

- After filtering, approximately 6K valid image-object pairs were retained. For each pair type, an **80/20 train-test split** was applied while maintaining distributional balance over object categories and depth separations.

- Each question is posed as: *"Which object is closer to the camera, the van (highlighted in red) or the cyclist (highlighted in blue)? Choose one: 1. red, 2. blue."*

## B EXPERIMENT DETAILS

### B.1 HYPERPARAMETER DETAILS

To ensure reproducibility and fairness, we carefully followed the official TinyLLaVA hyperparameter recommendations for stages 1 and 2, maintaining both the global batch size and learning rate as

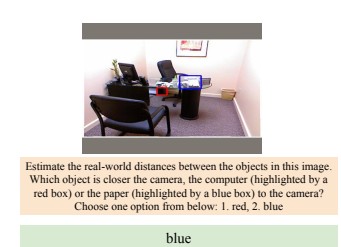 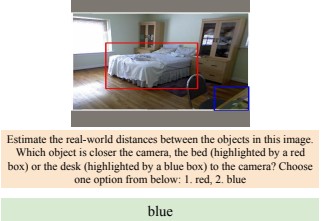 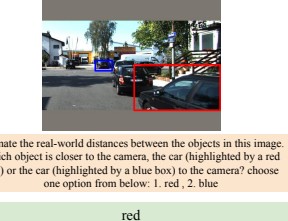

Figure 20: Examples of Relative Depth AVA Samples.

| Task | Model | lr 1e-5 | lr 1e-4 | lr 5e-4 | LoRA 64 | LoRA 128 | LoRA 256 |
|------|-------|---------|---------|---------|---------|----------|----------|
| **OCR** | DINOv2 | 7.26 | 9.97 | 10.6 | 10.85 | 9.97 | 10.98 |
| | SigLIP-2 | 79.68 | 81.18 | 77.63 | 81.51 | 81.18 | 81.25 |
| | CLIP | 54.23 | 60.44 | 60.9 | 60.79 | 60.44 | 61.73 |
| **Recognition** | DINOv2 | 83.92 | 86.31 | unstable | 86.39 | 86.31 | 86.46 |
| | SigLIP-2 | 87.04 | 88.19 | unstable | 88.42 | 88.19 | 88.36 |
| | CLIP | 83.08 | 85.02 | unstable | 84.79 | 85.02 | 84.18 |

Table 3: Hyperparameter exploration for OCR and Recognition tasks using three representative VFMs. Results are reported across different learning rates and LoRA dimensions.

prescribed (in Table 4.). For stage 3, which incorporates LoRA-based fine-tuning, we selected a learning rate of 1e-4 and explored multiple LoRA dimensions (64, 128, 256). To validate these choices, we conducted preliminary experiments using three representative VFMs: DINOv2, CLIP, and SigLIP-2. We evaluated performance on two representative AVA tasks (OCR and Recognition), as summarized in Table 3.

The results consistently show that the recommended learning rate of 1e-4 yields stable and strong performance, whereas alternative learning rates often underperform or lead to instability. Similarly, LoRA dimensions between 64 and 128 produce comparable and reliable results, while extreme values show diminishing returns. Based on these observations, we adopt the recommended configuration (learning rate 1e-4, LoRA dimension 128) throughout our experiments. The overall hyperparameters of Stage-1 vision-language alignment pretraining, Stage-2 visual instruction tuning and Stage-3 AVA-BENCH evaluation are shown in Table 4.

| | **TinyLLaVa** | | **AVA-BENCH** |
|---|---|---|---|
| Hyperparameter | Stage 1 | Stage 2 | Stage 3 |
| batch size | 16 | 4 | 4 |
| grad accum steps | 4 | 8 | 1 |
| LR | 1e-3 | 2e-5 | 1e-4 |
| LR schedule | cosine decay | | cosine decay |
| LR warmup ratio | 0.03 | | 0.03 |
| weight decay | 0 | | 0 |
| epoch | 1 | | 10 (20 for localization AVA) |
| optimizer | AdamW | | AdamW |
| DeepSpeed stage | 3 | | 3 |
| components finetuned | Connector | Connector + LLM | Connector + LoRA on LLM |
| sample size | 558K | 665K | – |

Table 4: Hyperparameters of TinyLLaVa and AVA-BENCH Evaluation Stage

### B.2 METRIC DETAILS

**Color Recognition.**    We use CIEDE2000 (Luo et al., 2001) to calculate the color differences using the `colour` Python library. Specifically, we convert the predictions and ground-truths from CIE XYZ tristimulus format to CIE L*a*b* colour space with `colour.XYZ_to_Lab`, followed by the `colour.delta_E(pred, gt, method="CIE 2000")` for color differences.

**Absolute Depth & Counting.**    We use the mean absolute error relative to the ground-truth (see Equation 1). This normalization ensures that errors involving greater distances or counts, which are inherently more challenging, are proportionally penalized less severely.

$$\text{MAE/GT} \;=\; \frac{1}{N}\sum_{i=1}^{N} \frac{\big|\, y_i - \hat{y}_i \,\big|}{y_i}. \tag{1}$$

where N is the number of test samples, $y_i$ is the ground-truth (depth or count) for sample $i$ and $\hat{y}_i$ the model prediction.

**Localization.**    We use the Generalized Intersection-over-Union (GIoU (Rezatofighi et al., 2019), Equation 2):

$$\text{GIoU}(A, B) = \frac{|A \cap B|}{|A \cup B|} \;-\; \frac{\big|\, C \setminus (A \cup B) \,\big|}{|C|}. \tag{2}$$

where $A$ and $B$ are the prediction and ground-truth bounding boxes and $C$ is the smallest (axis–aligned) enclosing box of $A \cup B$.

**OCR.**    We evaluate OCR performance with Average Normalized Levenshtein Similarity (ANLS) (Biten et al., 2019):

$$
\begin{aligned}
\text{NLS}(p, g) &= 1 - \frac{\text{Lev}(p, g)}{\max\big(|p|, |g|\big)}, \\
\text{ANLS} &= \frac{1}{N}\sum_{i=1}^{N} \text{NLS}(p_i, g_i) = \frac{1}{N}\sum_{i=1}^{N}\left(1 - \frac{\text{Lev}(p_i, g_i)}{\max\big(|p_i|, |g_i|\big)}\right).
\end{aligned}
\tag{3}
$$

where $\text{Lev}(p, g)$ is the (Levenshtein) edit distance and $|.|$ denotes string length and N is the number of testing samples.

**Others.**    All other AVAs employ standard accuracy metrics.

## C   MORE RESULTS AND ANALYSIS

### C.1   DETAILED OVERALL RESULTS

The results used for plotting Figure 7 and Figure 8 are presented in Table 5 where the best performance for each is AVA is **bold** and the second best is in *italics*. For each VFM, the first row is the performance, and the second row is the rank.

### C.2   DETAILED ANALYSES FOR EACH AVA

In the main results, we reported the overall performance of VFMs across various AVAs. However, aggregate metrics can sometimes obscure important nuanced insights. To gain deeper understanding, we conduct detailed analyses by partitioning test samples based on specific criteria (e.g., object size in localization tasks) and examining whether these subgroup trends align with overall performance. Generally, the detailed analyses affirm overall trends, but notable exceptions exist, particularly in localization.

| AVA | Abs. Depth | Rel. Depth | OCR | Counting | Localization | Object | Fine-grained | Scene | Action | Spatial | Emotion | Orientation | Texture | Color | Average |
|---|---|---|---|---|---|---|---|---|---|---|---|---|---|---|---|
| Metric | MAE/GT↓ | ACC↑ | ANLS↑ | MAE/GT↓ | GIOU↑ | ACC↑ | ACC↑ | ACC↑ | ACC↑ | ACC↑ | ACC↑ | ACC↑ | ACC↑ | CIEDE2000↓ | Ranking |
| SigLIP-2 | 0.0843 | *97.38* | **81.18** | **0.225** | **0.6738** | **88.19** | 90.17 | 72.60 | *45.33* | 99.50 | 58.39 | 79.2 | *94.08* | *12.25* | 2.4 |
|  | 3 | 2 | 1 | 1 | 1 | 1 | 3 | 3 | 2 | 2 | 5 | 4 | 2 | 2 |  |
| AIMv2 | 0.1008 | 95.71 | 62.44 | 0.254 | 0.5896 | 85.04 | **91.78** | 72.86 | 43.61 | 99.13 | **60.68** | 79.71 | **94.11** | 19.61 | 3.9 |
|  | 9 | 8 | 3 | 3 | 5 | 4 | 1 | 2 | 3 | 6 | 1 | 3 | 1 | 6 |  |
| SigLIP-1 | 0.0953 | 96.58 | *80.3* | *0.229* | 0.6103 | *87.84* | 90.94 | **73.4** | **45.76** | 99.07 | 59.6 | 78.91 | 93.66 | 12.64 | 3.6 |
|  | 6 | 7 | 2 | 2 | 4 | 2 | 2 | 1 | 1 | 7 | 4 | 5 | 3 | 3 |  |
| CLIP | 0.08461 | 97.04 | 60.44 | 0.290 | 0.5787 | 85.02 | 86.83 | 72.32 | 41.59 | 99.25 | *60.42* | 77.77 | 93.25 | 19.85 | 4.9 |
|  | 4 | 4 | 6 | 7 | 7 | 5 | 4 | 5 | 4 | 5 | 2 | 6 | 5 | 7 |  |
| InternVL-2.5 | *0.08212* | 97.00 | 60.88 | 0.269 | 0.5850 | 83.19 | 72.83 | 71.63 | 36.13 | 99.5 | 59.99 | 75.71 | 91.21 | 20.09 | 5.4 |
|  | 2 | 5 | 5 | 5 | 6 | 7 | 7 | 6 | 7 | 4 | 3 | 7 | 7 | 8 |  |
| RADIOv2.1 | **0.07645** | **97.92** | 62.44 | 0.257 | *0.6617* | 84.90 | 85.44 | 72.50 | 38.77 | **99.69** | 56.59 | *83.71* | 93.32 | 17.16 | 3.6 |
|  | 1 | 1 | 4 | 4 | 2 | 6 | 6 | 4 | 5 | 1 | 6 | 2 | 4 | 5 |  |
| DINOv2 | 0.08469 | 97.25 | 9.97 | 0.272 | 0.6598 | 86.31 | 85.5 | 70.99 | 37.45 | 99.50 | 54.44 | **85.54** | 93.06 | 21.54 | 5.3 |
|  | 5 | 3 | 7 | 6 | 3 | 3 | 5 | 7 | 6 | 3 | 7 | 1 | 6 | 9 |  |
| SAM | 0.09792 | 94.13 | 9.79 | 0.313 | 0.5216 | 76.68 | 40.06 | 58.28 | 17.25 | 90.22 | 36.88 | 69.37 | 81.02 | **9.87** | 7.8 |
|  | 8 | 9 | 8 | 8 | 8 | 8 | 8 | 9 | 9 | 8 | 9 | 8 | 9 | 1 |  |
| MiDas-3.0 | 0.09563 | 96.63 | 7.72 | 0.336 | 0.4490 | 75.05 | 32.83 | 60.19 | 18.25 | 53.58 | 40.40 | 67.37 | 86.38 | 13.28 | 8 |
|  | 7 | 6 | 9 | 9 | 9 | 9 | 9 | 8 | 8 | 9 | 8 | 9 | 8 | 4 |  |

Table 5: The detailed overall results where the best performance for each is AVA is **bold** and the second best is in *italics*. For each VFM, the first row is the performance, and the second row is the rank. Arrows indicate whether lower (↓) or higher (↑) is better.

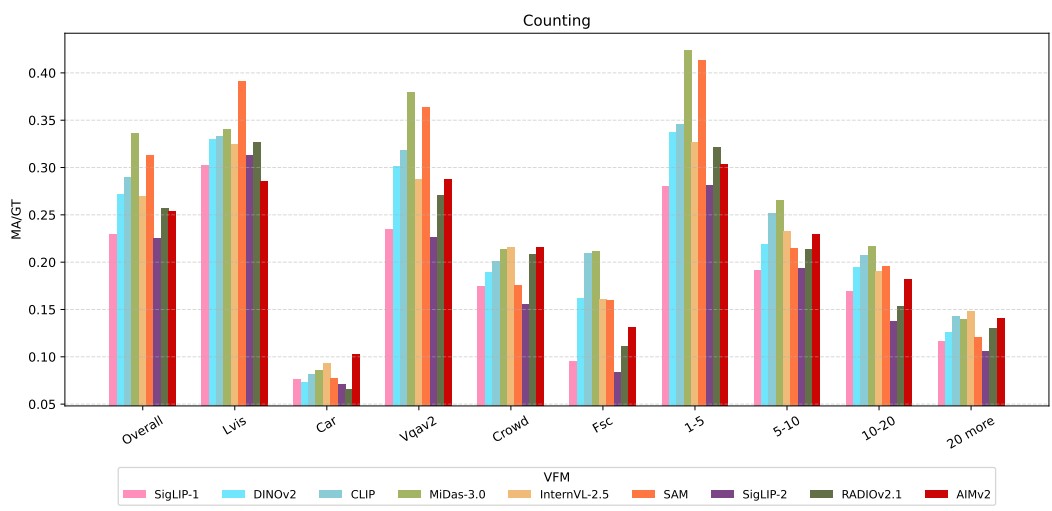

Figure 21: Detail results for counting for overall and different splits based on datasets and ground-truth count range. Lower MAE/GT is better.

**Localization.** We split localization testing samples based on normalized bounding box sizes (relative to image size), where 0.1 indicates an object occupies 10% of the image area. As illustrated in Figure 8 (b), VFMs surprisingly exhibit minimal performance differences when localizing large objects (0.3–0.5). Conversely, performance disparities amplify as object size decreases, revealing significant weaknesses in MiDas and SAM for smaller objects. Consequently, the lower overall performance of MiDas and SAM is predominantly due to poor handling of small targets. Practitioners should thus consider object size distributions when selecting VFMs; SAM and MiDas remain viable if target objects are predominantly large .

**Counting.** Counting performance is generally consistent across different datasets and count ranges (Figure 21). A notable exception is SAM, whose accuracy notably improves in denser scenarios.

**Emotion.** Emotion recognition results exhibit remarkable consistency, with rankings and relative performances highly stable across emotion categories (see Figure 22).

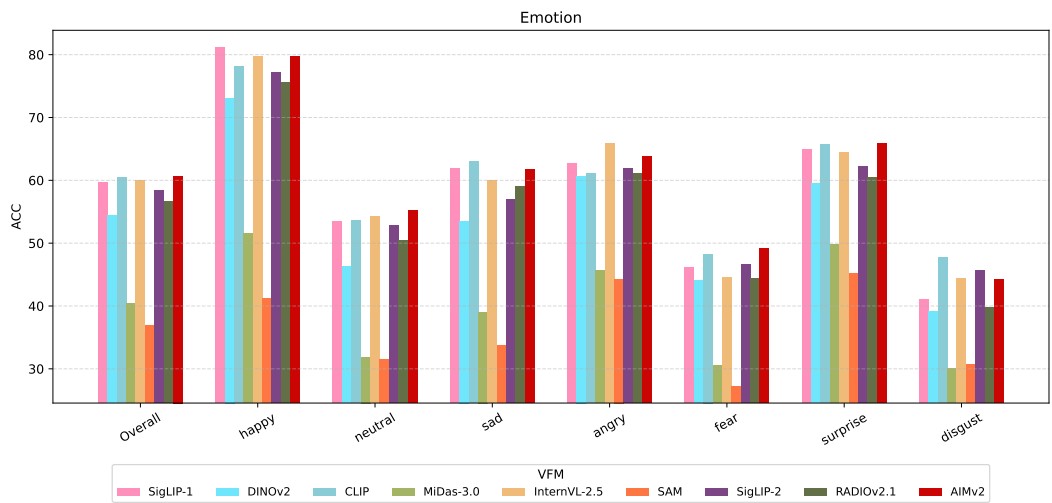

Figure 22: Detail results for emotion for overall and different splits based on emotion types.

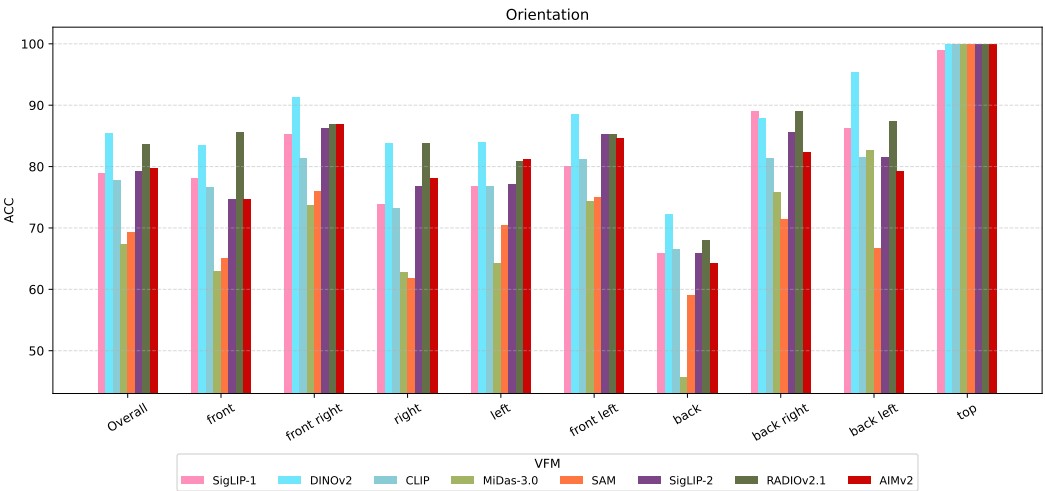

Figure 23: Detail results for orientation for overall and different splits based on viewpoint directions.

**Orientation.** Orientation performance remains consistent overall, with some intriguing exceptions. Specifically, VFMs universally achieve near-perfect accuracy for top-view images, presumably due to the distinctive nature of this viewpoint compared to side or frontal views (see Figure 23).

**Absolute Depth.** Overall, absolute depth performance is stable, though specific VFMs exhibit distinctive trends. SAM notably struggles with near objects but improves significantly with increased distance. Conversely, RADIO demonstrates an opposite pattern, excelling with nearer objects but deteriorating with greater distances (see Figure 24).

**Relative Depth.** Relative depth estimation shows uniformly high performance across VFMs, consistently surpassing 90% accuracy. SAM, however, underperforms notably in interior scenes, consistent with the earlier observation in absolute depth and counting that SAM handles smaller, exterior objects better (see Figure 25).

**Fine-grained Classification.** Fine-grained classification results are consistently robust across datasets, with the exception of SAM and MiDas, both of which are known to lack semantically rich features Espinosa et al. (2024); Chen et al. (2023), resulting in poorer performance (see Figure 26).

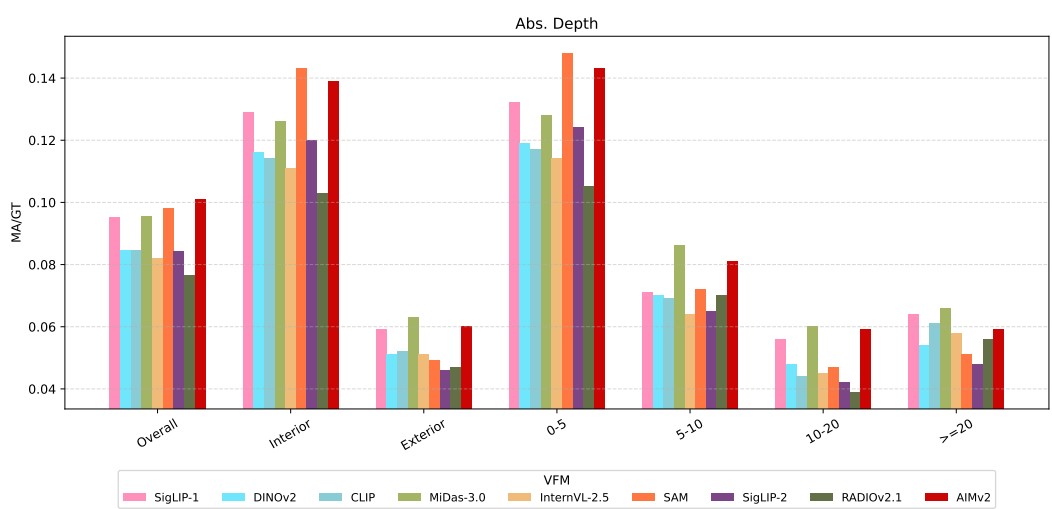

Figure 24: Detail results for absolute depth for overall and different splits based on scene type and object count range. Lower MAE/GT is better.

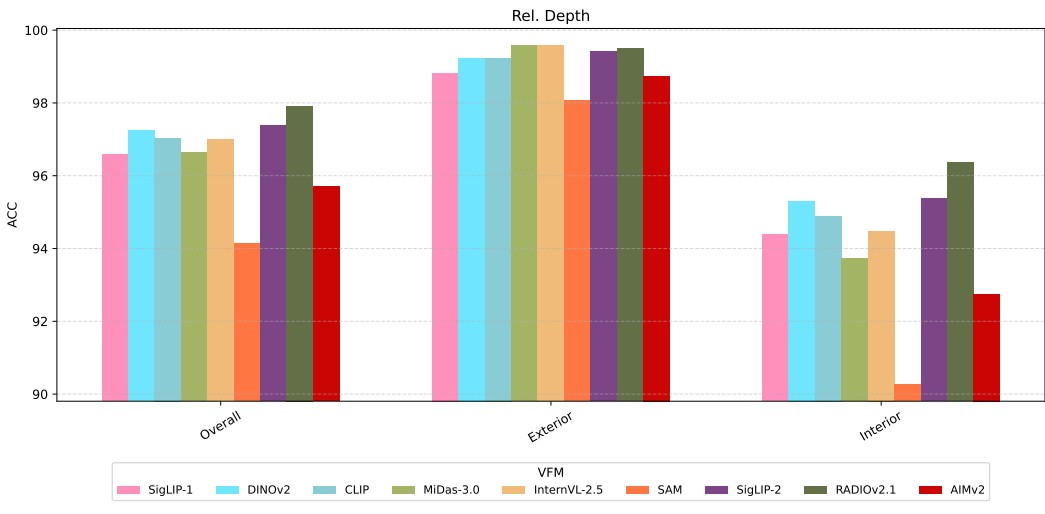

Figure 25: Detail results for relative depth for overall and different splits based on scene type.

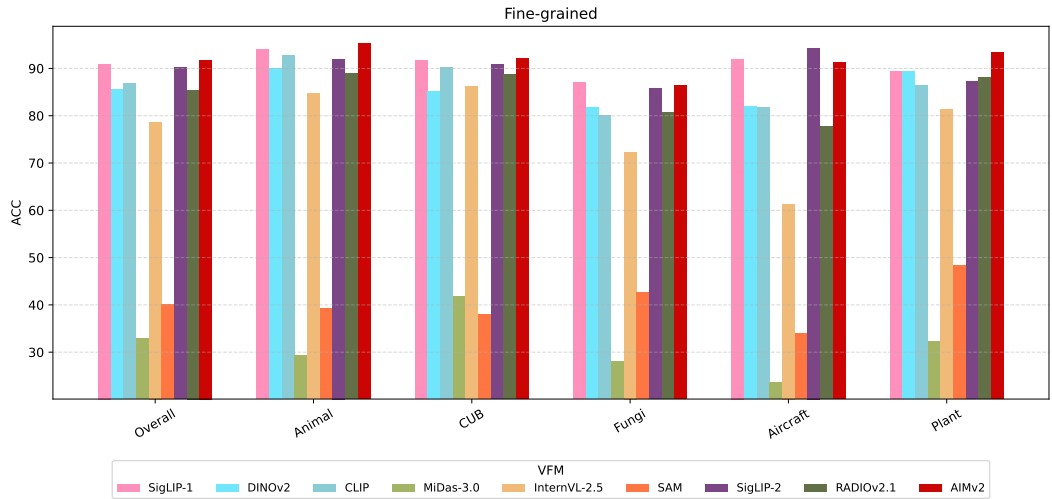

Figure 26: Detail results for fine-grained for overall and different splits based on dataset type.

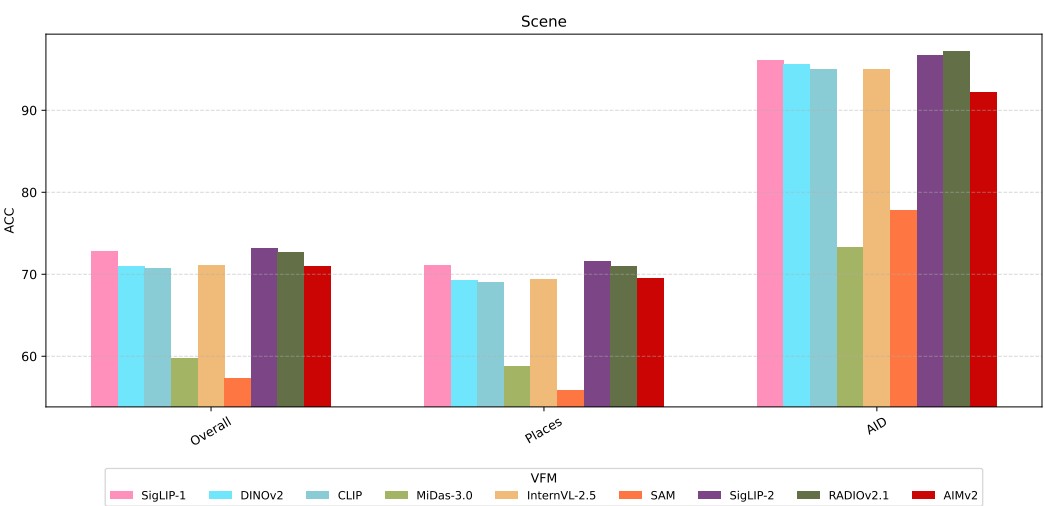

Figure 27: Detail results for scene for overall and different splits based on datasets.

**Scene Recognition.** Scene recognition performance is uniformly consistent across all evaluated datasets, echoing the patterns observed in fine-grained classification, where SAM and MiDas again lag behind other VFMs (see Figure 27).

**OCR.** OCR results show consistent patterns with those reported in Section 5.2, highlighting significant underperformance by non-language-aligned VFMs, such as DINOv2 and SAM. Notably, we observe that relative performances across VFMs are stable on short texts (length $< 20$). However, performance for CLIP and AIM sharply declines with longer text sequences (length $> 20$), indicating potential limitations in handling extensive textual information (see Figure 28).

# D    VFM DETAILS

Table 6 provides a detailed overview of the vision foundation models (VFMs) evaluated in our study. For each model, we list its architecture, parameter count, and training data, and we further summarize the training methodology in terms of supervision type, process, and loss functions.

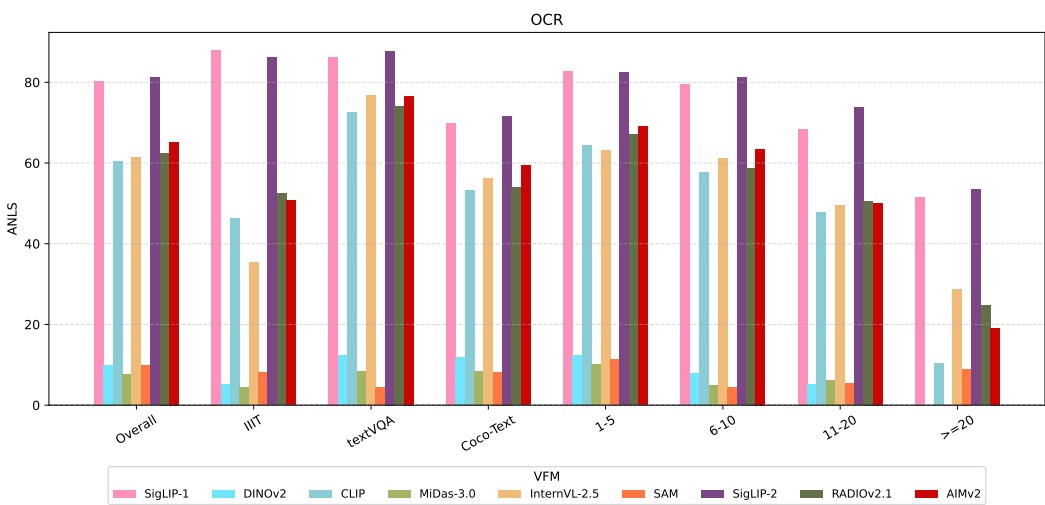

Figure 28: Detail results for counting for overall and different splits based on dataset domain and character length. Higher ANLS is better.

| VFM | Architecture | Parameters | Training Data | Training Details |
|---|---|---|---|---|
| SigLIP-2 | Dual-tower ViT encoders for image and text embeddings with MAP pooling layers | So400m (400M) | WebLI dataset | •Supervision: Supervised (image-text pairs) •Process: Pretrained on 40B samples, large-batch training (32k) •Loss: Pairwise sigmoid ITC + captioning/grounding |
| AIMv2 | ViT-based vision encoder with an autoregressive multimodal decoder for patch and token reconstruction | AIMv2-Huge (600M) | DFN, COYO, HQITP | •Supervision: Supervised (multimodal autoregressive) •Process: Pretrain (224 px) → finetune (336/448 px), long training •Loss: Joint reconstruction of image patches and text tokens |
| SigLIP-1 | Dual-encoder with independent ViT and text transformer projecting to a shared embedding space | So400m (400M) | WebLI dataset | •Supervision: Supervised (image-text pairs) •Process: Large-scale pretraining, efficient setup with 32k batch •Loss: Pairwise sigmoid contrastive loss |
| CLIP | Dual-tower model using a ViT image encoder and Transformer text encoder for contrastive alignment | ViT-L/14 (428M) | Internet-collected dataset | •Supervision: Supervised (image-text pairs) •Process: Weeks-long pretraining on multi-GPU/TPU with large batches (≥32k) •Loss: Contrastive InfoNCE |
| InternVL-2.5 | Large multimodal architecture combining a high-capacity ViT encoder with an LLM for image-text reasoning | InternVL2.5 (304M) | FaceCaption, GQA, ChartQA, Many other datasets | •Supervision: Supervised (multimodal LLM) •Process: Pretrained and finetuned on diverse datasets •Loss: Autoregressive next-token + alignment losses |
| RADIO v2.1 | ViT backbone with conditional positional encoding and multi-teacher feature distillation layers | RADIO-Huge (653M) | DataComp1B dataset | •Supervision: Supervised (teacher-student distillation) •Process: 600k steps, AdamW (WD=1e-4), batch scaling law (eff. BS 1024) •Loss: Multi-teacher distillation from CLIP, DINOv2, SAM-H |
| DINOv2 | ViT backbone with patch embedding and projection heads for self-supervised feature representation learning | ViT-Large (300M) | LVD-142M dataset | •Supervision: Self-supervised (no labels) •Process: Teacher-student distillation pipeline with deduplication and retrieval •Loss: Self-distillation contrastive objective |
| SAM | MAE-pretrained ViT-H image encoder paired with a prompt encoder handling points, boxes, masks, and text queries | ViT-H (637M) | SA-1B dataset | •Supervision: Supervised (segmentation masks) •Process: Pretrained encoder with MAE, promptable training on SA-1B •Loss: Segmentation mask prediction loss |
| MiDaS-3.0 | Multi-scale ResNet-based encoder-decoder network designed for monocular depth prediction from single images | ResNet-Encoder (123M) | DIML Indoor, MegaDepth, ReDWeb, WSVD | •Supervision: Supervised (depth ground truth) •Process: Multi-dataset pretraining, 60 epochs, Adam optimizer with different LR for new vs pretrained layers •Loss: Trimmed MAE (20%) + gradient regularizer |

Table 6: Details of Vision foundation model (VFM) used, including architecture, parameter scale, training data, and training procedures.

# E   RELATED WORKS

## E.1   VFM EVALUATION

Existing evaluation VFM protocols generally fall into two categories. The first focuses on task-specific capabilities, typically attaching tailored heads to VFMs, followed by fine-tuning and evaluation on dedicated datasets such as ImageNet for classification (Han et al., 2022) and COCO for detection or segmentation (Thisanke et al., 2023). For example, DINOv2 (Oquab et al., 2023) is evaluated on image and video classification, instance recognition, image retrieval, semantic segmentation, and depth estimation. For each task, a task-specific head is trained.

To better capture the diverse and complex perception challenges of the real world, recent studies advocate a more generic approach that leverages large language models (LLMs) as general-purpose heads, evaluating VFMs on broad Visual Question Answering (VQA) benchmarks (Liu et al., 2023; Zhu et al., 2023; Chowdhery et al., 2023). For example, in addition to the traditional task-specific evaluation, AIMv2 (Fini et al., 2024) and RADIO (Ranzinger et al., 2024) follow the LLM-based evaluation and use a Llama-3.0(8B) (Grattafiori et al., 2024) and a Vicuna-1.5(7B) (Zheng et al., 2023) respectively, demonstrating a shift towards generalized multimodal evaluation.

## E.2   ATOMIC VISUAL ABILITIES

As discussed in subsection A.1, foundational visual concepts—such as number, color, texture, object identity, and spatial relations—have long been recognized as crucial building blocks in compositional Text-to-Image (T2I) benchmarks. Given their foundational role in generation tasks, these primitives naturally underpin perceptual tasks as well. For example, a concept like 'number' directly translates into the perceptual task of counting.

A recent work (Chae et al.) introduced AVSBench to evaluate whether MLLMs understand basic *geometric* features, including angle, boundary, orthogonality, and curvature, which they refer to as atomic visual skills. However, AVSBench primarily targets geometric comprehension abilities required for geometric diagrams arising in high-school level mathematics. Moreover, AVSBench provides only test data for MLLMs without addressing potential mismatches between training and test data distributions—an issue highlighted in Section 2.2. Consequently, mispredictions in AVSBench evaluations may result from data distribution mismatches rather than genuine visual deficiencies in VFM. In contrast, AVA-BENCH explicitly emphasizes atomic visual abilities essential for general visual reasoning tasks commonly encountered in real-world scenarios. By aligning training and evaluation data distributions, AVA-BENCH ensures that evaluation outcomes reliably reflect genuine visual perceptual capabilities of VFMs.

Additionally, a concurrent work (Wu et al., 2025b) defines a set of atomic visual capabilities analogous to ours. However, their goal fundamentally differs from ours: (Wu et al., 2025b) aims to build a visual compositional tuning data recipe that builds complex capabilities from simple atomic capabilities, which can significantly reduce instruction-tuning data volume while maintaining strong performance. In contrast, AVA-BENCH's objective is to systematically evaluate VFMs against atomic visual abilities, pinpointing their exact strengths and weaknesses, and providing a comprehensive diagnostic tool to advance the continual development (Mai et al., 2022; Lomonaco et al., 2022; Mai et al., 2021; Shim et al., 2021; Mai et al., 2020) of robust vision foundation models.

# F   EVALUATION EFFICIENCY

An important advantage of our framework is its efficiency compared to prior LLM-based evaluation protocols. As summarized in Table 7, existing methods typically rely on large language models such as Vicuna-7B and require ≈230 A100 GPU hours per vision foundation model (VFM). By contrast, our approach adopts a lightweight 0.5B LLM and smaller training data (1.2M samples in total for stages 1 and 2), which reduces the cost to ≈28 A100 GPU hours while still preserving consistent and reliable VFM rankings. This design choice enables practical scaling to a wide range of models without incurring prohibitive resource demands.

For stage 3, our framework further leverages LoRA-based fine-tuning, where each AVA is trained on only 6K–10K samples. This procedure requires ≈4 A100 GPU hours per AVA on average, making it

| Protocol | LLM size | Stage 1&2 data | Stage 1&2 cost | Stage 3 |
|----------|----------|----------------|----------------|---------|
| Baseline [1] | Vicuna-7B | 1.9M | $\approx$230 A100 h | n/a |
| AVA-BENCH | Qwen2-0.5B+LoRa(stage 3) | 1.2M | $\approx$28 A100 h | Each AVA: avg 4 A100 h |

Table 7: Evaluation Efficiency Table

| Dataset | Copyright | License |
|---------|-----------|---------|
| Object365 | Objects365 Consortium | CC By 4.0 |
| LVIS | LVIS Consortium | CC By 4.0 |
| iNaturalist-2021 | iNaturalist (Terms of Service) | MIT |
| DIOR | N/A | N/A |
| VQAv2 | VQA Consortium | CC BY 4.0 |
| FSC | CVLab at StonyBrook | MIT |
| CARPK | Original image owners (PUCPR/PKLot) | N/A |
| Crowd Surveillance Dataset | N/A | N/A |
| CUB-200-2011 | Annotations: Catherine Wah et al.; images: original owners | CC0 (Public Domain) |
| FGVC-Aircraft | Annotations: S. Maji et al.; images: original owners | Research only (Non-commercial) |
| KITTI | Andreas Geiger, Philip Lenz, Christoph Stiller, Raquel Urtasun | CC BY-NC-SA 3.0 |
| NYU-DepthV2 | N/A | N/A |
| coco-text | SE(3) Computer Vision Group, Cornell Tech | CC BY 4.0 |
| IIIT5K | IIIT Hyderabad (annotations); images: original owners | N/A |
| TextVQA | VQA Consortium | CC BY 4.0 |
| EgoOrientBench | N/A | N/A |
| CURE-OR | OLIVES at Georgia Institute of Technology | MIT |
| Moment_int_time | Moments in Time authors | Research/Educational only |
| DTD | N/A | N/A |
| KTH-TIPS | N/A | N/A |
| KTH-TIPS2 | N/A | N/A |
| Places365 | MIT CSAIL, Bolei Zhou; images: original owners | MIT (code); images original copyright owners |
| AID | AID authors (Gui-Song Xia et al.); images from Google Earth providers | N/A |
| RAF-DB | N/A | N/A |
| ExpW | N/A | N/A |

Table 8: Dataset copyright and licensing information for all datasets used in AVA-BENCH

highly lightweight compared to full model finetuning. In summary, AVA-BENCH achieves a more diagnostic evaluation with considerably lower overhead than prior work.

## G  DATASET COPYRIGHT/LICENSE

To ensure ethical and legal use of datasets, we summarize the copyright and licensing information of all benchmarks employed in our experiments (Table 8). The majority of the datasets we use are publicly available under open licenses such as CC BY 4.0, MIT, or CC0, which permit research and redistribution with proper attribution. Some datasets (e.g., FGVC-Aircraft, Moments in Time) are restricted to research-only or educational use, and we adhered to these conditions. For datasets without explicit licensing details, we used them strictly within the scope of non-commercial academic research.

