# OpenReview forum: "AVA-Bench: Atomic Visual Ability Benchmark for Vision Foundation Models"
_ICLR.cc/2026/Conference — ICLR 2026 Conference Withdrawn Submission_

### Official Review · Reviewer_2KnY · 2025-10-26

**Soundness:** 2
**Presentation:** 3
**Contribution:** 2
**Rating:** 4
**Confidence:** 2

**Summary:**

The paper introduces AVA-Bench, an innovative and practical benchmark designed to evaluate the Atomic Visual Abilities (AVAs) of Visual Foundation Models (VFMs). This tool fills a key gap in existing evaluation methods by decomposing complex visual tasks into fundamental abilities, providing a clear view of VFMs’ strengths and weaknesses. The study is well-designed, and its findings offer valuable guidance for improving future VFMs.

**Strengths:**

1.	The paper first proposes a novel evaluation approach that decomposes visual tasks into Atomic Visual Abilities (AVAs).
2.	AVA-Bench covers 14 AVAs, ranging from low-level texture recognition to high-level spatial reasoning and emotion recognition, offering a broad evaluation scope. The dataset is constructed with attention to diversity and balance, ensuring reliable and fair assessment results.
3.	By revealing the performance of different VFMs across various AVAs, the benchmark helps MLLMs select the most suitable VFM based on specific task requirements.

**Weaknesses:**

Non-language-aligned VFMs (e.g., DINOv2, SAM) may suffer from visual information loss during alignment with language modalities, causing evaluation results to underrepresent their true visual capabilities. For example, in fine-grained recognition tasks, DINOv2’s accuracy drops sharply from 66.3% (pre-connector) to 25.67% (post-connector), indicating that the alignment process can degrade critical visual representations.

**Questions:**

1. It remains unclear whether VFMs possess genuine reasoning abilities or can contribute meaningfully to reasoning-related tasks.
2. AVA-Bench relies entirely on an LLM as a universal prediction head. Although results show that a smaller 0.5B LLM (Qwen2) can replace a 7B LLM (Vicuna-1.5) to reduce cost, the scaling behavior and impact on VLM performance remain insufficiently explored.
3. The paper should more clearly explain the differences between this work and general VLM benchmarks, such as MMMU and MMStar, which would help highlight and strengthen its contributions.

---

### Official Review · Reviewer_MTRY · 2025-10-31

**Soundness:** 3
**Presentation:** 3
**Contribution:** 3
**Rating:** 6
**Confidence:** 3

**Summary:**

This paper addresses a prevalent and critical problem in the evaluation of Vision Foundation Models (VFMs). Current evaluation methods typically rely on pairing VFMs with Large Language Models (LLMs) and testing them on general-purpose Visual Question Answering (VQA) benchmarks. The authors astutely identify two key blind spots in this approach: Data Distribution Mismatch and Ability Entanglement. To address these issues, the paper introduces AVA-BENCH, an innovative benchmark that explicitly disentangles complex visual tasks into 14 Atomic Visual Abilities (AVAs), such as localization, counting, depth estimation, spatial understanding, OCR, and more.

**Strengths:**

1. The problem tackled—how to fairly, transparently, and diagnostically evaluate VFMs—is a central challenge in the current multimodal and computer vision fields. The authors' analysis of the two blind spots in existing VQA evaluation is spot-on.

2. The core idea of AVA-BENCH ("ability decomposition") is excellent. Decomposing complex visual tasks into 14 atomic abilities and providing independent, distribution-matched datasets for each is a highly systematic and diagnostic approach. It shifts VFM evaluation from "what score does it get?" to "what is it good at, and what is it bad at?".

3. The paper covers VFMs from diverse training paradigms (e.g., language-supervised, self-supervised, segmentation-supervised), making the experiments very comprehensive. The analysis (as seen in Figures 5 & 6) goes beyond reporting numbers to truly reveal the "ability fingerprints" and trade-offs between different VFMs, such as DINOv2's strength in vision-centric tasks versus SigLIP's versatility.

**Weaknesses:**

1. The paper selects 14 AVAs. Although the authors explain their origins in the appendix (based on literature and VQA analysis), the "completeness" or "orthogonality" of this set remains open for discussion. For instance, why these 14? Are other critical atomic abilities missing? A more in-depth discussion of the selection criteria for this set would make the paper more persuasive.

2. The 14 AVAs vary in their level of "atomicity." For example, "color recognition" or "texture recognition" are arguably more "atomic" than "action recognition" or "fine-grained recognition." The latter (like action recognition) may inherently involve multiple implicit abilities such as localization and pose estimation. The paper could more clearly discuss these differences in complexity and why these "more complex" tasks are still classified as "atomic."

Minor suggestion: The main paper lacks a whole table to present the detailed evaluation results.

**Questions:**

See above

---

### Official Review · Reviewer_st1M · 2025-10-31

**Soundness:** 4
**Presentation:** 4
**Contribution:** 2
**Rating:** 4
**Confidence:** 4

**Summary:**

This paper introduces AVA-BENCH, a novel benchmark aimed at evaluating Vision Foundation Models (VFMs) based on 14 Atomic Visual Abilities (AVAs), such as localization, counting, depth estimation, orientation, and others. Unlike traditional VQA-based evaluation, which conflates multiple abilities and suffers from dataset misalignment, AVA-BENCH isolates and evaluates each perceptual skill independently. The framework uses a lightweight LLM (0.5B) as the multimodal head and fine-tunes it with LoRA over VFM features to measure performance on each AVA.

The study evaluates a diverse suite of VFMs (DINOv2, CLIP, SigLIP, SAM, MiDaS, AIMv2, etc.), revealing model-specific strengths and weaknesses across abilities, offering actionable insights for model selection. A key finding is that smaller VFM-LLM combinations can preserve relative performance rankings, significantly reducing computation without sacrificing fidelity.

**Strengths:**

1. The benchmark isolates 14 atomic visual abilities from standard VQA tasks, enabling diagnostic-level insight into VFM performance.
2. It aggregates 218K image-question pairs curated from 26 datasets.
3. It reveals ability-specific performance "fingerprints" across diverse VFMs (e.g., SigLIP, AIMv2, DINOv2, SAM).
4. Demonstrates that 0.5B LLMs retain VFM ranking accuracy while cutting GPU cost by 8× compared to 7B models.

**Weaknesses:**

1. **Lack of Connector Ablations**: A core component of the AVA-BENCH protocol is the LLM-based evaluation pipeline that attaches a lightweight language model to each VFM via a connector, trained using LoRA. However, the paper does not provide a systematic exploration or justification of the connector design choices, such as the LoRA rank, or whether certain VFMs require different connector hyperparameters to retain their perceptual strengths post-alignment. Since AVA-BENCH aims to serve as a generalizable diagnostic tool across all VFMs, it is critical to establish that connector-induced artifacts do not bias the evaluation. Without ablations on connector design, it is unclear whether performance drops are due to inherent VFM limitations or insufficient preservation of the VFM’s features by the connector. This uncertainty weakens confidence in the benchmark's comparative fidelity. For AVA-BENCH to be replicable and broadly valid, connector components and their training strategies (e.g., LoRA rank, learning rate, frozen depth, etc.) must be validated and stress-tested across diverse VFMs. Otherwise, one risks conflating the alignment's shortcomings with actual perceptual deficiencies of VFMs, leading to misleading performance fingerprints.

2. **Lack of Correlation Analysis with Standard VQA Benchmarks**: While AVA-BENCH offers granular diagnostic insights by decoupling visual abilities into 14 atomic dimensions, the paper does not investigate how these isolated ability scores correlate with performance on more complex, real-world VQA benchmarks. This limits our understanding of whether strong performance on individual AVAs (e.g., spatial reasoning or OCR) meaningfully translates into improved results on holistic tasks that require composition of multiple abilities, such as counting objects with specific attributes or relating text to visual cues in context. Without demonstrating such correlations, it is unclear if AVA-BENCH is predictive of real-world VQA competence or merely measuring isolated skills in a vacuum.

3. **Similarity to Prior Work (Novelty Concerns):** A significant limitation of the paper lies in its conceptual and empirical overlap with prior diagnostic studies of VFMs. Notably, many of the main findings in AVA-BENCH—such as the superior general performance of SigLIP-based models, the consistent baseline competence of CLIP variants, or the strong vision-only performance of DINOv2—have been previously documented in Cambrian. This raises questions about the extent of empirical novelty provided by the benchmark beyond revalidating known results in a more fine-grained setup. How does AVA training compare to FastVLM [1]?

[1] Vasu, P. K. A., Faghri, F., Li, C. L., Koc, C., True, N., Antony, A., ... & Pouransari, H. (2025). Fastvlm: Efficient vision encoding for vision language models. In Proceedings of the Computer Vision and Pattern Recognition Conference (pp. 19769-19780).

**Questions:**

1. Have the authors evaluated VFMs using simpler non-LLM decoders (e.g., linear probes or task-specific MLPs) to check whether the LLM-based head is introducing bias or performance bottlenecks—especially for AVAs like texture, where language grounding is not required? If so, do any contradictory trends emerge compared to the LLM-based evaluation?

---

### Official Review · Reviewer_dfxC · 2025-11-02

**Soundness:** 3
**Presentation:** 3
**Contribution:** 3
**Rating:** 4
**Confidence:** 3

**Summary:**

This work introduces a benchmark that evaluates Vision Foundation Models (VFMs) through 14 Atomic Visual Abilities (AVAs) that serve as the building blocks of visual reasoning. Unlike traditional Visual Question Answering (VQA) benchmarks that mix multiple skills, AVA-Bench isolates each ability using over 218K curated samples from 26 datasets, providing clear diagnostic insight into model strengths and weaknesses.

Across leading VFMs, language-supervised models like SigLIP and AIMv2 show broad competence, while self-supervised models like DINOv2 excel in geometric perception. Most VFMs handle low-level tasks well, but struggle with specific abilities such as small-object localization or OCR. By quantifying these atomic abilities this work provides an effective evaluation and selection of VFMs for downstream multimodal systems.

**Strengths:**

This work introduces a framework for evaluating Vision Foundation Models (VFMs) at the level of atomic perception capabilities. Through the 14 Atomic Visual Abilities (AVAs), such as localization, spatial reasoning, and color recognition,  a diagnostic benchmark that can identify the exact source of a model’s strengths or weaknesses is provided.

**Weaknesses:**

The deficiencies of vision foundation models that this work points out are well known. That is, vision models often struggle with precisely perceiving basic information despite being able to perceive higher-level objects well.

Also, the 14 atomic visual abilities seem to have been chosen somewhat arbitrarily. As an example, "motion or temporal understanding" could conceivably be an entry in the list of atomic visual abilities, but it is not included. Also, it is questionable whether "emotion" is an "atomic" ability.

While it seems that this benchmark is measuring useful abilities, it is unclear whether this benchmark must exist. While it is true that existing VQG benchmarks often require multiple visual abilities, perhaps the existing benchmarks are sufficient to drive progress in VFM, and the need to have a disentangled "atomic" benchmark is not clear to me.

I believe a good benchmark should measure something that existing benchmarks do not, but existing VQA benchmarks measure a superset of abilities that this benchmark is measuring.

**Questions:**

There is another paper with a similar motivation that presents a dataset named "Atomic Visual Skill Bench" while this paper is titled "Atomic Visual Ability Bench", essentially the same name. When I first saw the title, I admittedly suspected possible plagiarism, but after reading the paper, I found no actual impropriety.

That said, using an almost identical name creates an appearance of impropriety, and I question whether this naming choice was prudent.

---

### Note · Authors · 2025-11-14

I have read and agree with the venue's withdrawal policy on behalf of myself and my co-authors.